# Multitask Learning with No Regret:
# from Improved Confidence Bounds to Active Learning

**Pier Giuseppe Sessa**[*]
ETH Zürich
piergiuseppe.sessa@inf.ethz.ch

**Pierre Laforgue**[*]
Università degli Studi di Milano
pierre.laforgue@unimi.it

**Nicolò Cesa-Bianchi**
Università degli Studi di Milano
Politecnico di Milano
nicolo.cesa-bianchi@unimi.it

**Andreas Krause**
ETH Zürich
krausea@ethz.ch

## Abstract

Multitask learning is a powerful framework that enables one to simultaneously learn multiple related tasks by sharing information between them. Quantifying uncertainty in the estimated tasks is of pivotal importance for many downstream applications, such as online or active learning. In this work, we provide novel confidence intervals for multitask regression in the challenging agnostic setting, i.e., when neither the similarity between tasks nor the tasks' features are available to the learner. The obtained intervals do not require i.i.d. data and can be directly applied to bound the regret in online learning. Through a refined analysis of the multitask information gain, we obtain new regret guarantees that, depending on a task similarity parameter, can significantly improve over treating tasks independently. We further propose a novel online learning algorithm that achieves such improved regret without knowing this parameter in advance, i.e., automatically adapting to task similarity. As a second key application of our results, we introduce a novel multitask active learning setup where several tasks must be simultaneously optimized, but only one of them can be queried for feedback by the learner at each round. For this problem, we design a no-regret algorithm that uses our confidence intervals to decide which task should be queried. Finally, we empirically validate our bounds and algorithms on synthetic and real-world (drug discovery) data.

## 1 Introduction

In many real-world applications, one often faces multiple related tasks to be solved sequentially or simultaneously. The goal of multitask learning (MTL) [4] is to leverage the similarities across the tasks to obtain more accurate and robust models. Indeed, by jointly learning multiple tasks, MTL can exploit their statistical dependencies, yielding better generalization and faster learning than treating each task independently. MTL has gained significant attention in recent years, as it has been shown to be effective in a wide range of applications, including natural language processing, computer vision, federated learning, and drug discovery, see e.g., [11, 19, 16, 27].

A very natural model for learning across multiple tasks is the agnostic multitask (MT) regression approach of [13]. This utilizes a multitask kernel that can interpolate between running $N$ (number of tasks) independent regressions, and regressing all tasks to their common average, depending on a tunable parameter. Notably, such a kernel does not require any knowledge neither about tasks' features

---

[*]equal contribution

37th Conference on Neural Information Processing Systems (NeurIPS 2023).

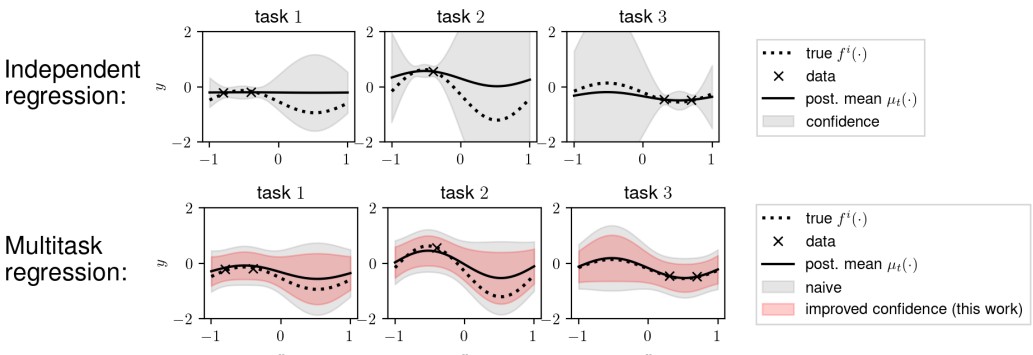

Figure 1: Independent vs. Multitask (MT) regression. MT regression leverages data coming from multiple related tasks and can yield more accurate and more confident estimates. In this work, we show naive confidence intervals are overly conservative and provide improved ones (shaded in red).

nor about their similarity, thus finding good application in several domains. For instance, Cavallanti et al. [5] study it for online classification, and Cesa-Bianchi et al. [6] for online convex optimization.

However, it is much less understood how to *quantify the uncertainty* of such MT regression, i.e., assessing confidence in the estimated tasks. In particular, as also outlined by [13] as an open problem, it is important to assess their generalization error as a function of the kernel parameter. Appropriately characterizing these confidence intervals is indeed of crucial importance for a whole set of downstream applications. More concretely, multitask confidence intervals are used in online learning to inform the next decision to be made [6]. In active learning—as we show next—these intervals are pivotal to deciding the most informative task to query.

In this work, we study the agnostic MT regression setup of [13], and provide *new multitask confidence intervals* (see Figure 1 for a visualization) for the full range of the kernel parameter. Our intervals hold in the so-called adaptive setting, i.e., without requiring i.i.d. data, and are *tighter up to a $\sqrt{N}$ factor* than the naive ones employed in [6]. Moreover, we provide the first bounds for the information gain of MT regression and utilize them—together with the derived intervals—to obtain *tighter online learning guarantees*. The latter depend on a task similarity parameter and can significantly improve over treating tasks independently. Additionally, we propose an adaptive no-regret algorithm that exploits task similarity without knowing this parameter in advance. Finally, we consider a novel multitask *active learning* setup, where tasks should be simultaneously optimized but only one of them can be queried at each round. We show that the newly derived intervals are also crucial in such a setting, and provide a new algorithm that ensures sublinear regret. We demonstrate the superiority of the derived intervals over previously proposed algorithms on synthetic as well as real-world drug discovery tasks.

**Related work.** The agnostic MT regression approach of [13] reduces the learning of $N$ tasks to a single regression problem, as a function of the MT kernel parameter. When combined with support vector machines, it was shown effective in a series of classification problems [13, 24], and since then was studied in various further settings. Cavallanti et al. [5], e.g., analyze mistake bounds for online MT classification algorithms as a function of the kernel parameter. Cesa-Bianchi et al. [6], instead, utilize the MT kernel to prove regret bounds in online MT learning with bandit feedback. Inspired by this, [10] focuses on learning more general kernel structures from data. An important question not addressed by previous work, though, is how to properly quantify the uncertainty of the obtained task estimates. This problem is well-understood in single-task learning (e.g., [2, 25, 8]) but remains largely unexplored in MT domains. As shown in [6], MT confidence intervals can in principle be obtained by a naive application of the single-task guarantees of [2]. However, as we show in Section 2, the so-obtained intervals are extremely conservative and—as a result—can hamper the MT learning performance. Our intervals are tighter by a factor up to $\sqrt{N}$ w.r.t. the naive ones from [6], yielding novel online learning regret guarantees which can provably improve over treating tasks independently.

Compared to MT online learning [5, 6], where a single task is revealed to the learner at each round, a series of works have considered learning multiple tasks *simultaneously*, i.e., taking a decision for each one of them. Dekel et al. [12], e.g., propose the use of a shared loss function to account for tasks' relatedness, Lugosi et al. [20] studies the computational tractability of taking multiple actions with joint constraints, while Cavallanti et al. [5] propose a matrix-based extension of the multitask Perceptron algorithm. In all of these works, however, the learner receives feedback from *all* the

tasks. In Section 4, instead, we focus on the challenging setup where only one task can be queried by the learner at each round. Hence, in addition to choosing good actions, the learner faces the *active learning* task of assessing the most informative feedback, in order to achieve sublinear regret. Perhaps most related to ours, is the offline contextual Bayesian optimization setup of [7, 18] where the goal is to compute the best strategy for each context (task) with minimal function interactions. However, unlike us, [7, 18] do not guarantee sublinear regret but provide only sample-complexity results.

Finally, we note that MT confidence intervals and regret guarantees were also recently derived by [9], albeit in a different setup and regression model. Indeed, the authors of [9] focus on multi-objective optimization where they ought to learn *multi-output* functions (each output corresponding to a task) using matrix-valued kernels. Although their setup can be related to ours, it crucially requires all tasks to be observed at each round, leading to different challenges than ours, see Appendix A.1 for details.

**Notation.** We use $[N] \coloneqq \{1, \ldots, N\}$, $\mathbb{1}_N$ for the vector in $\mathbb{R}^N$ full of ones. Norms of functions are always taken w.r.t. the natural RKHS norm, so that we drop the subscript for simplicity of writing.

## 2 Improved Confidence Intervals for Multitask Kernel Regression

In this section, we introduce the MT kernel regression setting, and prove our refined confidence intervals. Of independent interest, these results are then leveraged in Sections 3 and 4 to derive novel regret bounds for online and active multitask learning. All proofs are deferred to the Appendix.

### 2.1 Multitask Kernel Regression

Given an input space $\mathcal{X}$, equipped with a (single task) scalar kernel $k_\mathcal{X} : \mathcal{X} \times \mathcal{X} \to \mathbb{R}$, the goal of MT kernel regression is to jointly learn $N$ different functions $f_1, \ldots, f_N$ from $\mathcal{X}$ to $\mathbb{R}$, all belonging to $\mathcal{H}_{k_\mathcal{X}}$, the RKHS associated to $k_\mathcal{X}$. To do so, the learner is given a set of triplets $\{(i_s, x_s), y_s\}_{s=1}^t$ consisting of a measured task index $i_s \in [N]$, a measured point $x_s \in \mathcal{X}$, and a noisy measurement $y_s = f_{i_s}(x_s) + \xi_s$, where $\xi_s$ is an independent random variable to be specified later. We can further define the multitask function $f^{\mathrm{mt}} : [N] \times \mathcal{X} \to \mathbb{R}$ such that $f^{\mathrm{mt}}(i, \cdot) = f_i$, and the multitask kernel

$$k\big((i, x), (i', x')\big) = k_\mathcal{T}(i, i') \cdot k_\mathcal{X}(x, x') \,, \tag{1}$$

where $k_\mathcal{T}$ is a kernel on the tasks. In certain cases, the latter might be given as input to the learner, either under the form of the task Gram matrix, or via task features and an assumed (e.g., linear) similarity [17]. However, in practice such information is usually not accessible to the learner. In such a case, a standard *agnostic* approach to MT regression [5, 6, 10, 13, 24] then consists in leveraging a parameterized task kernel of the form

$$k_\mathcal{T}(i, i') = \big[K_{\mathrm{task}}(b)\big]_{ii'} \,, \quad \text{with} \quad K_{\mathrm{task}}(b) = \frac{1}{1+b} I_N + \frac{b}{1+b} \frac{\mathbb{1}_N \mathbb{1}_N^\top}{N} \in \mathbb{R}^{N \times N} \,. \tag{2}$$

Intuitively, parameter $b \geq 0$ governs how similar the tasks are thought to be. When $b = 0$, we have $K_{\mathrm{task}}(b) = I_N$, such that $k_\mathcal{T}(i, i') = \delta_{ii'}$, and the tasks are considered to be independent. When $b$ goes to $+\infty$, we have $K_{\mathrm{task}}(b) = \mathbb{1}_N \mathbb{1}_N^\top / N$, and all tasks are considered to be one single common task. Any choice of $b \in (0, +\infty)$ corresponds to a tradeoff between these two regimes. We make this intuition explicit in Proposition 2 (Appendix A.1). Note that all quantities depending on the kernel do by definition depend on $b$. We use the notation $| b$ to make this dependence explicit when relevant.

Given a history of measurements $\{(i_s, x_s), y_s\}_{s=1}^t$, one may then estimate $f^{\mathrm{mt}}$, or equivalently the $\{f_i\}_{i=1}^N$, by standard kernel Ridge regression using the MT kernel $k$. One obtains the estimates

$$\mu_t(i, x \,|\, b) = \boldsymbol{k}_t(i, x)^\top \big(K_t + \lambda I_t\big)^{-1} \boldsymbol{y}_{1:t} \,, \tag{3}$$

$$\sigma_t^2(i, x \,|\, b) = k\big((i, x), (i, x)\big) - \boldsymbol{k}_t(i, x)^\top \big(K_t + \lambda I_t\big)^{-1} \boldsymbol{k}_t(i, x) \,, \tag{4}$$

where $\boldsymbol{k}_t(i, x) = \big[k\big((i_s, x_s), (i, x)\big)\big]_{s=1}^t$, $K_t = \big[k\big((i_s, x_s), (i_{s'}, x_{s'})\big)\big]_{s,s'=1}^t$, $\boldsymbol{y}_{1:t} = \big[y_s\big]_{s=1}^t$, and $\lambda > 0$ is some regularization parameter. Functions $\mu_t$ and $\sigma_t^2$ can be interpreted as the posterior mean and variance of a corresponding Gaussian Process model, see [25, 8]. In the next section, we will utilize $\mu_t$ and $\sigma_t^2$ to construct high-probability confidence intervals for the multitask function $f^{\mathrm{mt}}$.

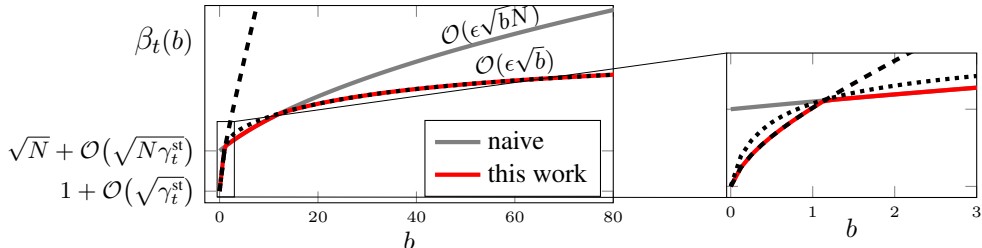

Figure 2: Novel multi-task confidence width $\beta_t^{\text{new}}(b)$ (see Theorem 1) visualized for large and small values of $b$. It improves over the naive width $\beta_t^{\text{naive}}(b)$ by a factor of $\sqrt{N}$ at $b = 0$ and as $b \to +\infty$. Problem parameters were set to $B = 1, \epsilon = 0.4, N = 20, t = 4$, and $\gamma_t^{\text{mt}}(b) = \gamma_t^{\text{st}} = 0$ for all $b$.

**Information gain.** An important quantity when analyzing (multitask) kernel regression is the so-called *(multitask) information gain*:

$$\gamma_T^{\text{mt}}(b) = \frac{1}{2} \ln \left| I_T + \lambda^{-1} K_T \right|.$$

It can be interpreted as the reduction in uncertainty about $f^{\text{mt}}$ after having observed a given set of $T$ datapoints. Similarly to single-task setups [25, 8], we use $\gamma_T^{\text{mt}}$ in the next sections to characterize our confidence intervals and regret bounds. Note that $\gamma_T^{\text{mt}}$ depends on the multitask kernel through $K_T$, and hence on $b$. In Section 3, we exploit the properties of our multitask kernel to obtain a sharper control over $\gamma_T^{\text{mt}}$, which is then fundamental to derive improved regret bounds.

## 2.2 Improved Confidence Intervals

In this section, we utilize the regression estimates obtained in Equations (3) and (4) to construct high probability confidence intervals around the unknown multitask function $f^{\text{mt}}$. First, we assume that $\|f_i\| \le B$ for all $i \in [N]$, as it is standard in single-task regression. Moreover, let $f_{\text{avg}} = (1/N) \sum_{i=1}^{N} f_i$ be the average task function, and define

$$\epsilon = \max_i \|f_i - f_{\text{avg}}\|/B. \tag{5}$$

Note that by definition $\epsilon \in [0, 2]$. The smaller $\epsilon$, the more similar the tasks are, the limit case being that all tasks are equal, attained at $\epsilon = 0$. At the other extreme, when $\epsilon \gg 0$ tasks are highly distant and ought to be learned independently. The deviation $\epsilon$ plays a crucial role in the subsequent analysis.

**A naive confidence interval.** As discussed in [6], it is possible to construct the multitask feature map $\widetilde{\psi}$ associated to $k$. One may then rewrite $f^{\text{mt}}(i, x) = \langle \widetilde{f}, \widetilde{\psi}(i, x) \rangle$, where $\widetilde{f}$ is a transformed version of $f^{\text{mt}}$ which satisfies $\|\widetilde{f}\| \le B\sqrt{N(1 + b\epsilon^2)}$, see Appendices A.1 and A.2 for details. MT regression thus boils down to single-task regression, over the modified features $\widetilde{\psi}(i, x)$, and with target function $\widetilde{f}$. One can then employ well-known linear regression results to obtain confidence intervals for $f^{\text{mt}}$. Using [1, Theorem 3.11, Remark 3.13] and the definition of $\gamma_t^{\text{mt}}(b)$, with probability $1 - \delta$ we have that for all $t \in \mathbb{N}, i \in [N]$, and $x \in \mathcal{X}$ it holds $\left| \mu_t(i, x \mid b) - f^{\text{mt}}(i, x) \right| \le \beta_t^{\text{naive}}(b) \cdot \sigma_t(i, x \mid b)$, where

$$\beta_t^{\text{naive}}(b) = B\sqrt{N(1 + b\epsilon^2)} + \lambda^{-1/2} \sqrt{2 \left( \gamma_t^{\text{mt}}(b) + \ln(1/\delta) \right)}.$$

Note that the above confidence interval was already established in [6, Theorem 1]. As expected, it depends on $B$, $N$, $b$, and in a decreasing fashion with respect to $\epsilon$. However, we argue that the above naive choice can be *extremely conservative*. Indeed, when $b = 0$, MT regression treats tasks independently, see Proposition 2. Hence, a valid confidence width from [2, 1, 8] is $\mathcal{O}\left(B + \sqrt{\gamma_t^{\text{st}}}\right)$, where $\gamma_t^{\text{st}}$ is the single-task maximum information gain. Instead, noting that $\gamma_t^{\text{mt}}(0) = \mathcal{O}\left(N\gamma_t^{\text{st}}\right)$, see Proposition 1, the naive choice provides $\beta_t^{\text{naive}}(0) = \sqrt{N} \cdot \mathcal{O}\left(B + \sqrt{\gamma_t^{\text{st}}}\right)$, which is larger by a factor $\sqrt{N}$. A similar suboptimality gap of $\sqrt{N}$ can also be proven when $b$ tends to $+\infty$. Motivated by the above observation, we derive a novel confidence width that is less conservative than $\beta_t^{\text{naive}}(b)$ for the whole range of possible kernel parameters $b$.

**Theorem 1** (Multitask confidence intervals). *Let $f^{\text{mt}} \colon [N] \times \mathcal{X} \to \mathbb{R}$ such that for all $i \in [N]$, $f_i := f^{\text{mt}}(i, \cdot)$ belongs to the RKHS associated to $k_{\mathcal{X}}$ and $\|f_i\| \le B$. Moreover, let $\mu_t$ and $\sigma_t$ be the*

*regression estimates of Equations* (3) *and* (4) *with task kernel* $k_\mathcal{T}(i,j) = [K_{\text{task}}(b)]_{ij}$, *parameter* $\lambda \in [1/(1+b), 1]$, *and noise* $\{\xi_\tau\}_{\tau=1}^t$ *i.i.d.* 1-*sub-Gaussian. Then, with probability at least* $1 - 2\delta$,

$$\left| \mu_t(i, x \mid b) - f^{\text{mt}}(i, x) \right| \le \beta_t^{\text{new}}(b) \cdot \sigma_t(i, x \mid b), \qquad \forall\, t \in \mathbb{N}, i \in [N], x \in \mathcal{X}\,,$$

*where* $\quad \beta_t^{\text{new}}(b) = \min\left\{ \beta_t^{\text{naive}}(b),\, \beta_t^{\text{small-}b}(b),\, \beta_t^{\text{large-}b}(b) \right\},$

$$\beta_t^{\text{small-}b}(b) = B(1 + b\epsilon)\sqrt{\frac{1 + bN}{1 + b}} + \lambda^{-1/2}\sqrt{2(1 + bN)\big(\gamma_t^{\text{st}} + \ln(N/\delta)\big)}\,,$$

$$\beta_t^{\text{large-}b}(b) = B\sqrt{\frac{(1 + b\epsilon)^2}{1 + b} + \frac{2bN}{1 + b} + \frac{2b(1 + b\epsilon)^2}{N\lambda^2(1 + b)^3} t^2} + \lambda^{-1/2}\sqrt{2\big(\gamma_t^{\text{mt}}(b) + \ln(1/\delta)\big)}\,.$$

The obtained improved confidence width $\beta_t^{\text{new}}(b)$ is the minimum between three confidence widths, see Figure 2. The first one is the naive one $\beta_t^{\text{naive}}(b)$, obtained by standard arguments as outlined above, while $\beta_t^{\text{small-}b}(b)$ and $\beta_t^{\text{large-}b}(b)$ (dashed and dotted lines in Figure 2) are novel and useful for small and large values of $b$, respectively. Indeed, note that we have $\beta_t^{\text{small-}b}(b) \xrightarrow{b \to 0} \mathcal{O}\big(B + \sqrt{\gamma_t^{\text{st}}}\big)$, which is the expected single-task confidence width and $\sqrt{N}$ smaller than $\beta_t^{\text{naive}}(0)$. Similarly, as $b$ goes to $+\infty$ we have $\beta_t^{\text{large-}b}(b) \overset{b \to +\infty}{\sim} \mathcal{O}\big(B\sqrt{b\epsilon^2 + 2N + 2\epsilon^2 t^2/N}\big) \overset{b \to +\infty}{\sim} \mathcal{O}\big(\epsilon B\sqrt{b}\big)$, while $\beta_t^{\text{naive}}(b) \overset{b \to +\infty}{\sim} \mathcal{O}\big(\epsilon B\sqrt{Nb}\big)$. The obtained confidence width is therefore always smaller than the naive one, but also tighter by a factor $\sqrt{N}$ for the extreme choices $b = 0$ and $b = +\infty$.

From a technical viewpoint, $\beta_t^{\text{small-}b}$ and $\beta_t^{\text{large-}b}$ are obtained by viewing MT regression as a single-task regression over the inflated features $\widetilde{\psi}(i, x)$, as also done in [6]. However, unlike [6], we explicitly leverage the expressions of $\widetilde{\psi}(i, x \mid b)$ and $K_{\text{task}}(b)$ as functions of $b$. In particular, because of the structure of $K_{\text{task}}$, the regression kernel matrix is a rank-one perturbation of a block diagonal matrix, a fact that we exploit, e.g., via Lemma 2. Moreover, we note that refined widths can be obtained if one has access to task-specific constants $B_i$ and $\epsilon_i$. For simplicity of exposition, we focus on uniform (over tasks) $B$ and $\epsilon$. Also, a tighter data-dependent $\beta_t^{\text{large-}b}$ can be utilized as outlined in Appendix A.2.2. Finally, we remark that the obtained multitask intervals do not require i.i.d. data and thus apply to the *adaptive design* setting where data are, e.g., sequentially acquired by the learner, as shown in the next section.

## 3   New Guarantees for Multitask Online Learning

In this section, we show how the improved confidence interval established in Theorem 1 can be used to derive sharp regret guarantees for multitask online learning. To do so, we also prove novel bounds for the multitask information gain $\gamma_T^{\text{mt}}(b)$. For $t = 1, 2, \ldots$ the learning protocol is as follows: nature reveals task index $i_t \in [N]$; the learner chooses strategy $x_t \in \mathcal{X}$ and pays $f^{\text{mt}}(i_t, x_t)$; the learner observes the noisy feedback $y_t = f^{\text{mt}}(i_t, x_t) + \xi_t$. The goal is to minimize for any horizon $T$ the multitask regret

$$R^{\text{mt}}(T) = \sum_{t=1}^T \max_{x \in \mathcal{X}} f^{\text{mt}}(i_t, x) - \sum_{t=1}^T f^{\text{mt}}(i_t, x_t)\,. \tag{6}$$

In the next subsection, we provide a generic algorithm to minimize (6). In particular, we show that naive choices of parameters allow to recover previous approaches with their guarantees, while using the refined confidence width $\beta_t^{\text{new}}(b)$ derived in Theorem 1 yields significant improvements.

### 3.1   Algorithm and regret guarantees

In line with the online learning literature, our approach is based on the multitask Upper Confidence Bound, defined for any $t \in \mathbb{N}$ as

$$\text{ucb}_t(i, x \mid b) = \mu_t(i, x \mid b) + \beta_t(b) \cdot \sigma_t(i, x \mid b)\,. \tag{7}$$

Here $\beta_t \colon \mathbb{R}_+ \to \mathbb{R}_+$ is a function which assigns a confidence width $\beta_t(b)$ to each kernel parameter $b$. We consider the general strategy MT-UCB (see Algorithm 1) which, at each round $t$ selects

**Algorithm 1** `MT-UCB`

**Require:** Domain $\mathcal{X}$, kernel $k_{\mathcal{X}}$, number of tasks $N$, width functions $\{\beta_t\}_{t\in\mathbb{N}}$, parameters $b, \lambda$.
1: **for** t = 1, 2, … **do**
2:   Observe $i_t$,
3:   Play $x_t = \arg\max_{x\in\mathcal{X}} \mathrm{ucb}_{t-1}(i_t, x \,|\, b)$,
4:   Observe $y_t = f^{\mathrm{mt}}(i_t, x_t) + \epsilon_t$,
5:   Update $\mathrm{ucb}_t(\cdot, \cdot \,|\, b)$ based on (3),(4), and (7).

| Algorithm | $\beta_t$ | $b$ | $\lambda$ |
|---|---|---|---|
| `IGP-UCB` [8] | $\beta_t^{\mathrm{small}\text{-}b}$ | 0 | 1 |
| `GoB.Lin` [6] | $\beta_t^{\mathrm{naive}}$ | $b$ | 1 |
| This work | $\beta_t^{\mathrm{new}}$ | $b$ | $\frac{N+b}{N+bN}$ |

Table 1: Recovering previous works by appropriate choices of $\beta_t$, $b$, and $\lambda$.

$x_t = \arg\max_{x\in\mathcal{X}} \mathrm{ucb}_{t-1}(i_t, x \,|\, b)$. As summarized in Table 1, both the strategy that runs $N$ independent instances of `IGP-UCB` (one for each task), and `GoB.Lin` from [6] are particular cases of `MT-UCB`. Importantly, whenever $\beta_t(b)$ is set such that $[\mu_t(\cdot, \cdot \,|\, b) \pm \beta_t(b) \cdot \sigma_t(\cdot, \cdot \,|\, b)]$ is a valid confidence interval for $f^{\mathrm{mt}}(\cdot, \cdot)$, the regret of `MT-UCB` can be controlled through the following lemma.

**Lemma 1.** *Suppose that $\lambda \geq (N + b)/(N + bN)$, and that for all tasks $i$, point $x$, and time $t$, we have $f^{\mathrm{mt}}(i, x \,|\, b) \in [\,\mu_t(i, x \,|\, b) \pm \beta_t(b) \cdot \sigma_t(i, x \,|\, b)\,]$. Then, the multitask regret of `MT-UCB` satisfies*

$$R^{\mathrm{mt}}(T) \leq 4\,\beta_T(b)\sqrt{\lambda\,T\gamma_T^{\mathrm{mt}}(b)}\,.$$

The main novelty of Lemma 1 is that the right-hand side scales with $\lambda^{1/2}$, which might be chosen smaller than 1. This improvement is due to the fact that multitask posterior variances are smaller than $(N + b)/(N + bN) \leq 1$. The right-hand side also depends on the multitask information gain $\gamma_T^{\mathrm{mt}}(b)$, which is nontrivial to compute or upper bound. In the next proposition, we provide practical upper bounds of $\gamma_T^{\mathrm{mt}}(b)$, in terms of the kernel parameter $b$ and the single-task information gain $\gamma_T^{\mathrm{st}}$.

**Proposition 1.** *Let $\lambda \leq 1$, $N \geq 2$, and $T_i \geq 1$ for all $i \in [N]$. Then, for any $b \geq 0$, we have*

$$\gamma_T^{\mathrm{mt}}(b) \leq N\gamma_T^{\mathrm{st}} - \frac{Nb}{8(1+b)}, \qquad \text{and} \qquad \gamma_T^{\mathrm{mt}}(b) \leq \gamma_T^{\mathrm{st}} + \frac{T}{2\lambda(1+b)}\,.$$

We can now combine Theorem 1, Lemma 1, and Proposition 1 to obtain our main result: a bound on the multitask regret of `MT-UCB` run with the confidence width $\beta_t^{\mathrm{new}}$ from Theorem 1 and a specific $\lambda$.

**Theorem 2.** *Assume that $B \geq 1$, and that `MT-UCB` is run with $\beta_t = \beta_t^{\mathrm{new}}$ from Theorem 1, and $\lambda = (N + b)/(N + bN)$. Let $b = N/\epsilon^2$ if $T \leq N$, $b = 1/\epsilon^2$ if $T \geq N$ and $\epsilon \leq N^{-1/4}T^{-1/2}$, and $b = 0$ otherwise. Let $R^{\mathrm{st}}(T) = B\sqrt{T\gamma_T^{\mathrm{st}}} + \sqrt{T\gamma_T^{\mathrm{st}}\big(\gamma_T^{\mathrm{st}} + \ln(1/\delta)\big)}$ be the single task regret bound achieved by `IGP-UCB` (up to constant factors). Then, there exists a universal constant $C$ such that with probability $1 - 2\delta$ we have (up to $\log N$ factors)*

$$R^{\mathrm{mt}}(T) \leq C \min\left\{ \sqrt{N}R^{\mathrm{st}}(T) \,,\; R^{\mathrm{st}}(T) + \epsilon BT^{3/2}\Big(\sqrt{\gamma_T^{\mathrm{st}} + \ln(1/\delta)} + \epsilon\sqrt{T}\Big) \,, \right.$$
$$\left. R^{\mathrm{st}}(T) + \epsilon BT\sqrt{N}\Big(\sqrt{\gamma_T^{\mathrm{st}} + \ln(1/\delta)} + \epsilon\sqrt{NT}\Big) \right\}.$$

The regret bound of Theorem 2 is the minimum between three bounds, obtained exploiting the three different regimes of the confidence width $\beta_t^{\mathrm{new}}$ derived in Theorem 1 (see Figure 2). The **first bound** is obtained using $\beta_t^{\mathrm{new}} \leq \beta_t^{\mathrm{small}\text{-}b}$, and shows that our approach cannot be worse than independent learning. Indeed, it can be checked that, when facing $N$ tasks, the regret of running $N$ independent instances of `IGP-UCB` can be bounded by $\sqrt{N}$ times the single-task regret bound of `IGP-UCB`, that we denoted by $R^{\mathrm{st}}(T)$. Note however that our analysis slightly differs, insofar as we leverage the multitask information gain, while the independent analysis uses Jensen's inequality to aggregate the individual bounds, see Appendix B for details. Note finally that we are able to recover this bound as $\beta_t^{\mathrm{small}\text{-}b}$ is tight at $b = 0$, unlike $\beta_t^{\mathrm{naive}}$. The **second bound** uses $\beta_t^{\mathrm{new}} \leq \beta_t^{\mathrm{large}\text{-}b}$ and consists of two terms: the single task regret bound and an additional term that scales with the task deviation $\epsilon$. When the latter is small, i.e., when tasks are similar, the dominant term is $R^{\mathrm{st}}(T)$, as if only one task were solved. The **third bound** is similar, but obtained using $\beta_t^{\mathrm{new}} \leq \beta_t^{\mathrm{naive}}$ and is useful when $T \geq N$. In

contrast with the independent bound, which does not exploit the task structure, the last two bounds show that multitask learning is always beneficial when the horizon $T$ (and thus the additional $\epsilon$-related term) is small. As expected, this is particularly true when the number of tasks $N$ is large: while the independent bound increases, the second bound *does not depend on* $N$. On the other hand, one can note that the condition on $\epsilon$ to improve over independent becomes more constraining as the horizon $T$ increases. This suggests that the benefit of multitask may vanish with the number of available points per task, an observation which is well-known by practitioners, see e.g. [21]. As far as we know, this work is the first one to provide theoretical evidence of such a phenomenon in online MT learning.

We conclude this section by comparing Theorem 2 to existing results. As already mentioned in the above discussion, independent `IGP-UCB` is a particular case of `MT-UCB`, such that we cannot be worse than the independent approach. We incidentally recover its regret bound as the first bound in the minimum of Theorem 2. Regarding `GoB.Lin`, since it is also a specific instance of `MT-UCB` (for $\beta_t = \beta_t^{\mathrm{naive}}$ and $\lambda = 1$), Lemma 1 allows to recover its regret bound [6, Theorem 1].

**Corollary 1** (Regret of `GoB.Lin` [6]). *For any b, the multitask regret of `GoB.Lin` using parameter b satisfies with probability $1 - \delta$*

$$R^{\mathrm{mt}}(T) \leq 4\beta_T^{\mathrm{naive}}(b)\sqrt{T\gamma_T^{\mathrm{mt}}(b)} \leq 6\left(B\sqrt{N(1 + b\epsilon^2)} + \sqrt{\gamma_T^{\mathrm{mt}}(b) + \ln(1/\delta)}\right)\sqrt{T\gamma_T^{\mathrm{mt}}(b)}. \quad (8)$$

If tasks are similar, i.e., when $\epsilon \ll 1$, bound (8) suggests to choose $b > 0$; this does not impact too much the first term, but makes $\gamma_T^{\mathrm{mt}}(b)$ smaller. However, we recall that the above bound instantiated with $b = 0$ does not recover the independent bound. It is instead $\sqrt{N}$ bigger, since $\beta_t^{\mathrm{naive}}$ is not tight at $b = 0$. Hence, the `Gob.Lin` analysis is not sufficient to show that multitask learning improves over independent learning. Our refined analysis, which uses instead $\beta_t^{\mathrm{new}}$, closes this gap.

## 3.2 Adapting to unknown task similarity

In this section, we consider the case where parameter $\epsilon$ (i.e., a bound on the task deviation from the average, see (5)) is a-priori unknown. Despite this challenge, we show that the regret bound of Theorem 2 can be approximately attained using an adaptive procedure, `AdaMT-UCB` (Algorithm 3), relegated to Appendix B.4 due to space limitations. The proposed approach is inspired by the model selection scheme of [22, Section 7] with a few important modifications that we will outline at the end of this section. `AdaMT-UCB` considers a plausible set of parameters $\mathcal{E} = \{e_1, \ldots, e_{|\mathcal{E}|}\} \subset (0, 2]$ and, for each $e \in \mathcal{E}$, initializes an instance of the `MT-UCB` algorithm with parameters set according to Theorem 2 assuming $\epsilon = e$. We denote such an instance as `MT-UCB`$(e)$. Moreover, we use the notation $\mathrm{ucb}_t^e$ to denote the upper confidence bounds constructed by `MT-UCB`$(e)$. We assume the existence of some $e \in \mathcal{E}$ such that $e \geq \epsilon$, so that at least one of the learners is *well-specified* (i.e., its confidence bounds contain $f^{\mathrm{mt}}$ with high probability). Our goal is to incur a regret which grows as the regret of the learner with the smallest $e$ such that $e \geq \epsilon$, since the smaller the $e$ the smallest the regret bound (see Theorem 2), as long as $e$ is a valid upper bound for $\epsilon$. Let us identify with $e^\star$ such learner.

At each round $t$, `AdaMT-UCB` uses learner $e_t = \min \mathcal{E}$, and plays the action $x_t$ suggested by it, i.e., the maximizer of $\mathrm{ucb}_t^{e_t}(i_t, \cdot)$. Then, all `MT-UCB`$(e)$ learners are updated based on the observed reward. In the meantime, a *misspecification test* is carried out to check whether learner $e_t$ is well-specified. Such a test compares the obtained cumulative reward, a lower confidence estimate on such reward according to the other learners, and the believed regret of learner $e_t$. As long as the test does not trigger, the regret of learner $e_t$ is controlled by the believed one. Instead, if the test triggers, learner $e_t$ can be considered misspecified with high probability. As a result, it gets removed from $\mathcal{E}$ and a *new epoch* starts with the new set $\mathcal{E}$. Let $\overline{R_\star^{\mathrm{mt}}}(T)$ denote the regret bound (Theorem 2) of learner $e^\star$ had it been chosen from round 0. We can state the following.

**Theorem 3.** *Assume that there exists $e \in \mathcal{E}$ such that $e \geq \epsilon$, and let $M$ be the number of learners $e \in \mathcal{E}$ such that $e < \epsilon$ (i.e., the number of misspecified learners in $\mathcal{E}$). The regret of `AdaMT-UCB` satisfies with high probability $R^{\mathrm{mt}}(T) = \mathcal{O}\left(\sqrt{M + 1} \cdot \overline{R_\star^{\mathrm{mt}}}(T)\right)$.*

Clearly, the number $M$ of misspecified learners is not known in advance but is always less than $|\mathcal{E}|$. Note that when $\epsilon = 0$, we have $M = 0$ and we recover the single task regret bound. Moreover, given $\rho \leq 1$, we show in Appendix B.4 that one can attain a multiplicative accuracy $\rho$ over $\epsilon$, assuming that $\epsilon \geq \epsilon_{\min} > 0$, through an exponential grid with $M$ being polylogarithmic in $1/\rho$ and $1/\epsilon_{\min}$.

**Relation with the approach of [22].** Compared to [22, Section 7]—where the goal is to adapt to an unknown features' dimension—the set of learners considered in `AdaMT-UCB` share *the same dimension* $d$. This allows us to exploit the following two novelties with respect to [22]: (1) *all* learners are updated from the data gathered from learner $i_t$ (Line 6 in Algorithm 3), and (2) the lower confidence bounds $L^e$ in the misspecification test (Line 8) are all computed using action $x_t$ (i.e., the action recommended by learner $i_t$), as opposed to using the actions recommended by each learner $e$. Both these points are only applicable to our setting, leading to a simpler regret analysis.

## 4    Multitask Active Learning

The goal of the online learning setup of Section 3 is to optimize the tasks sequentially revealed by nature. In some situations (e.g., in [20] or the drug discovery problem considered in Section 5), however, we care about the performance of multiple tasks *simultaneously*, to eventually learn the best strategy for each one of them. Moreover, we ought to do so with minimal interactions $T$, i.e., minimizing the queries of the function $f^{\mathrm{mt}}$. We capture this by the following *active learning* protocol.

**Learning protocol and regret.** At each round $t$, the learner: chooses a strategy $\{x_t^i, i \in [N]\}$ *for each task*, chooses *which task* $i_t \in [N]$ to query, and observes the noisy feedback $y_t = f^{\mathrm{mt}}(i_t, x_t) + \xi_t$. The learner's goal is to minimize the *active learning* regret:

$$R_{\mathrm{AL}}^{\mathrm{mt}}(T) = \sum_{t=1}^{T} \frac{1}{N} \sum_{i=1}^{N} \max_{x \in \mathcal{X}} f^{\mathrm{mt}}(i, x) - \sum_{t=1}^{T} \frac{1}{N} \sum_{i=1}^{N} f^{\mathrm{mt}}(i, x_t^i).$$

Compared to the online learning regret of Equation (6), the learner's performance at each round is here measured by the average reward coming from *each* task (as opposed to just the task presented by nature). Moreover, compared to online learning, the learner faces the additional challenge of choosing—at each round—from which task information should be gathered. Intuitively, more difficult (or informative) tasks should be queried more often to ensure $R_{\mathrm{AL}}^{\mathrm{mt}}(T)$ grows sublinearly. To the best of our knowledge, the above protocol and regret notion are novel in the multitask literature.

In Algorithm 2 we present `MT-AL`, an efficient strategy that ensures sublinear active learning regret. Like in `MT-UCB`, `MT-AL` constructs confidence intervals around $f^{\mathrm{mt}}$ and, at each round, select strategy $x_t^i = \arg\max_{x \in \mathcal{X}} \mathrm{ucb}_{t-1}(i, x)$ for each task $i \in [N]$. When it comes to selecting which task to query, `MT-AL` selects $i_t \in \arg\max_{i \in [N]} \beta_{t-1}^i \sigma_{t-1}(i, x_t^i)$, i.e., the task for which the believed optimizer $x_t^i$ is subject to max-

| **Algorithm 2** `MT-AL` |
| :--- |
| **for** t=1,…, T **do** |
| $\quad x_t^i = \arg\max_{x \in \mathcal{X}} \mathrm{ucb}_{t-1}(i, x), \forall i \in [N]$ |
| $\quad i_t = \arg\max_{i \in [N]} \beta_{t-1}^i \sigma_{t-1}(i, x_t^i)$ |
| $\quad$ Observe: $y_t = f^{\mathrm{mt}}(i_t, x_t^{i_t}) + \xi_t$ |
| $\quad$ Update $\mathrm{ucb}_t(\cdot, \cdot)$ and $\sigma_t(\cdot, \cdot)$ based on |
| $\quad$ observations. |

imal uncertainty (we use generic task-dependent widths $\beta_t^i$ for completeness). This rule, also known as *uncertainty sampling* in the literature [23], intuitively makes sure the learner can control the regrets for the tasks not queried and leads to the following theorem.

**Theorem 4.** *Suppose that for all tasks $i$, point $x$, and time $t$, we have that $f^{\mathrm{mt}}(i, x) \in [\mu_t(i, x) \pm \beta_t^i \cdot \sigma_t(i, x)]$. Then, the* `MT-AL` *algorithm ensures the active learning regret is bounded by*

$$R_{\mathrm{AL}}^{\mathrm{mt}} \leq 2 \sum_{t=1}^{T} \beta_t^{i_t} \sigma_t(i_t, x_t^{i_t}),$$

*where $\{i_t\}$ is the sequence of queried tasks and $\{x_t^{i_t}\}$ the strategies selected for each of them.*

The above bound only relies on `MT-AL` utilizing valid intervals around $f^{\mathrm{mt}}$ and thus applies more broadly than our agnostic MT regression, e.g., when such intervals are constructed using a known multitask kernel $k\big((i, x), (i', x')\big)$. However, Theorem 4 shows the active learning regret heavily depends on the constructed intervals, similar to online learning. In `MT-AL`, these are additionally utilized for deciding which task to query at each round. When specialized to our agnostic MT kernel and improved confidence, we obtain the following.

**Corollary 2.** *Let* `MT-AL` *utilize the MT regression estimates of Eq. (3)-(4) with parameters set according to Theorem 2. Moreover, let $\overline{R^{\mathrm{mt}}}(T)$ be the bound on the online learning regret obtained in Theorem 2. Then, with high probability, we have $R_{\mathrm{AL}}^{\mathrm{mt}}(T) \leq \overline{R^{\mathrm{mt}}}(T)$.*

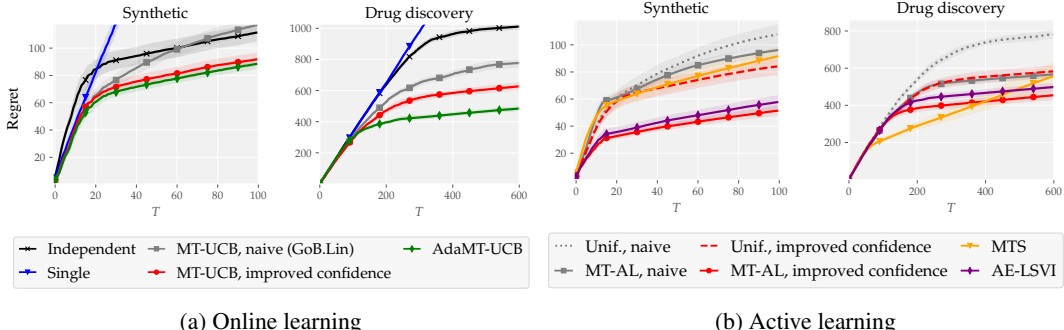

(a) Online learning  (b) Active learning

Figure 3: Online and active learning regrets on synthetic and drug discovery MHC-I data, respectively. When utilizing the improved confidence intervals, `MT-UCB` and `MT-AL` outperform the other baselines.

Thus, `MT-AL` ensures the active learning regret is always bounded by its online learning counterpart. Moreover, the same considerations as in Theorem 2 apply also here, regarding the benefit of multitask learning over independent single-task regression for instance.

# 5   Experiments

The goal of our experiments is to evaluate the effectiveness of the studied MT regression, and in particular of the improved confidence intervals obtained in Section 2, both in online learning and active learning setups. We utilize the following synthetic and real-world data[2].

*Synthetic data:* We generate tasks of the form $f_i = (1 - \delta) \cdot \bar{f} + \delta \cdot f_{\text{dev}}^i, i \in [N]$, where $\bar{f}, f_{\text{dev}}^i$ are random unit vectors representing a common model and individual deviations, respectively. Moreover, actions consist of $10^4$ vectors $x \in \mathbb{R}^d$ from the sphere of radius 10. Observation noise is unit normal.

*Drug discovery MHC-I data [27]:* The goal is to discover the peptides with maximal binding affinity to each Major Histocompatibility Complex class-I (MHC-I) allele. The dataset from [27] contains the standardized binding affinities ($IC_{50}$ values) of different peptide candidates to the MHC-I alleles (tasks). For each allele, the dataset contains $\sim 1000$ peptides represented as $x \in \mathbb{R}^{45}$ feature vectors. For our experiments, we utilize the 5 alleles A-$\{0201, 0202, 0203, 2301, 2402\}$, since they were shown in [27] to share binding similarity. Note that such a problem falls into our multitask active learning setup, since we would like to retrieve the best peptide for each allele minimizing the number of interactions (i.e., lab experiments). Nevertheless, we also consider its online learning analog where we care about finding the best peptides for each revealed allele.

**Online learning.** At each round $t$, a random task $i_t \in [N]$ is observed and point $x_t$ is selected according to the following baselines: (1) *Independent*, which runs $N$ independent `IGP-UCB` [8] algorithms (corresponding to `MT-UCB` with $b = 0$), (2) *Single*, which treats all tasks to be the same and runs a unique single-task `IGP-UCB` (corresponding to `MT-UCB` with $b = +\infty$), (3) `MT-UCB` which utilizes an appropriate parameter $0 < b < \infty$ as well as a bound on the tasks similarity $\epsilon$ (for synthetic data this can be exactly computed, while for MHC-I data we use $\epsilon = 0.3$) and utilizes the *naive* (i.e., `Gob.Lin`) or *improved* confidence bounds, and (4) `AdaMT-UCB` which is run with the same $b$ but uses the set of plausible deviations $\mathcal{E} = \{.1, .2, \dots, 1\}$ instead of knowing the true $\epsilon$. For choosing $b$, we sweep over possible values and select the best-performing one, keeping it fixed for all the baselines.

**Active learning.** We follow the multitask active learning setup of Section 4. All baselines utilize confidence intervals from the agnostic MT regression of Section 2, where $\epsilon$ and $b$ are chosen as for online learning. Moreover, they all utilize the improved confidence intervals, unless otherwise specified. We compare: (1) *Unif.* which chooses the task $i_t$ to be queried uniformly at random (but still selects $x_t^i \in \arg\max_x \text{ucb}_t^i(i, x)$) and employs the *naive* or the *improved* confidence intervals, the offline contextual Bayesian optimization baselines (2) `MTS` [7] and (3) `AE-LSVI` [18], and (4) `MT-AL` which utilizes the *naive* or the *improved* confidence intervals.

---

[2]code available at: https://github.com/sessap/multitask-noregret.

We report the cumulative regret (online and active learning, respectively) of the considered baselines in Figure 3, averaged over 5 runs. For the synthetic data, we report results for $d = 4, N = 5, \delta = 0.4$, but provide a full set of experiments for different parameters in Appendix D. In Appendix D we also report the frequencies of each task being queried in our active learning experiments. In Figure 3 (a), both `MT-UCB` and `AdaMT-UCB` lead to superior performance compared to the *Independent* and *Single* baselines, demonstrating the benefits of MT regression. In addition, the improved confidence intervals significantly outperform the naive ones. Moreover, we observe `AdaMT-UCB` achieves comparable (sometimes even better, see Appendix D) performance to `MT-UCB`. Indeed, instead of using a conservative choice of $\epsilon$, the misspecification test (Line 8 of Algorithm 3) of `AdaMT-UCB` allows to use a smaller $\epsilon$ and only increase it when there is evidence that the constructed intervals do not contain the true tasks. In active learning (Figure 3 (b)), we observe `MT-AL` has a significant advantage over the uniform sampling baselines and `MTS`, while performing comparably to `AE-LSVI` (both methods are similar as discussed in Appendix C.3). Moreover, its regret is bounded by the online learning regret of `MT-UCB`, conforming with Theorem 4. Importantly, the improved confidence intervals play a crucial role also here and enable a drastic performance improvement compared to the naive ones.

## 6 Future Directions

We believe this paper opens up several future research directions. The derived confidence intervals, as well as our analysis of the multitask information gain, heavily exploit the structure of the task Gram matrix $K_{\text{task}}(b)$, see Equation (2). However, it remains unclear whether these can be extended to more general kernels. According to the graph perspective of [13], $K_{\text{task}}(b)$ can be seen as $K_{\text{task}}(b) = I_N + L(b)$, where $L(b) \in \mathbb{R}^{N \times N}$ is the Laplacian matrix of a *clique* graph with vertices $[N]$ and edge weight $b$. Hence, it would be interesting to extend our results to different graph structures. Furthermore, we believe the proposed multitask confidence intervals hold potential for various related domains., e.g., to assess uncertainty in safety-critical systems [3], or to balance exploration-exploitation in multitask reinforcement learning [26], spam filtering [15], or personalized health [14]. In such applications, the introduced notion of active learning regret can serve as a measure of the overall sample efficiency.

## Acknowledgements

This work was partially supported by ELSA (European Lighthouse on Secure and Safe AI) funded by the European Union under grant agreement No. 101070617. PL and NCB gratefully acknowledge the financial support from the MUR PRIN grant 2022EKNE5K (Learning in Markets and Society), funded by the NextGenerationEU program within the PNRR scheme (M4C2, investment 1.1), the FAIR (Future Artificial Intelligence Research) project, funded by the NextGenerationEU program within the PNRR-PE-AI scheme (M4C2, investment 1.3, line on Artificial Intelligence), and the EU Horizon CL4-2022-HUMAN-02 research and innovation action under grant agreement 101120237, project ELIAS (European Lighthouse of AI for Sustainability).

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
