# Supplementary Material

We gather here the technical proofs and additional results complementing the main paper.

## A   Multitask Regression and Improved Confidence Intervals

**Notation.**   In order to coincide with the notation of [6], we introduce the matrix $A(b) \in \mathbb{R}^{N \times N}$ such that $A^{-1}(b) \coloneqq K_{\text{task}}(b)$. For all $b \geq 0$ we have

$$A(b) = (1+b)I_N - b\frac{\mathbb{1}_N \mathbb{1}_N^\top}{N}, \qquad A(b)^{1/2} = \sqrt{1+b}\,I_N + \left(1 - \sqrt{1+b}\right)\frac{\mathbb{1}_N \mathbb{1}_N^\top}{N},$$

$$A(b)^{-1} = \frac{1}{1+b}I_N + \frac{b}{1+b}\frac{\mathbb{1}_N \mathbb{1}_N^\top}{N}, \qquad A(b)^{-1/2} = \frac{1}{\sqrt{1+b}}I_N + \left(1 - \frac{1}{\sqrt{1+b}}\right)\frac{\mathbb{1}_N \mathbb{1}_N^\top}{N}.$$

Note that in the following we often drop the dependence in $b$ and use only $A$, $A^{-1}$, $A^{1/2}$, $A^{-1/2}$.

### A.1   Further intuitions about the MT regression of Section 2.1

Given the history of measurements $\{(i_s, x_s), y_s\}_{s=1}^t$, the agnostic MT regression considered in Section 2.1 estimates $f^{\text{mt}}$ (or equivalently the $f_i$) by standard kernel Ridge regression, i.e., by computing

$$\mu_t = \arg\min_{h \in \mathcal{H}_k} \sum_{s=1}^t \left(h(i_s, x_s) - y_s\right)^2 + \lambda\|h\|_{\mathcal{H}_k}^2, \tag{9}$$

where $\mathcal{H}_k$ is the RKHS associated to kernel $k$, and $\lambda > 0$ some regularization parameter. In the next proposition, we provide some intuition about the role of parameter $b$ by exhibiting regression problems that are equivalent to problem (9) when $b = 0$ and $b = +\infty$. We relegate its proof to the end of this section.

**Proposition 2.** *Solving the multitask kernel Ridge regression problem* (9) *for $b = 0$ is equivalent to solve $N$ independent kernel Ridge regressions, using each task's data separately, i.e., we have*

$$\forall\, i \in [N], \qquad \mu_t(i, \cdot \mid b = 0) = \arg\min_{h \in \mathcal{H}_{k_\mathcal{X}}} \sum_{s\,:\,i_s = i} \left(h(x_s) - y_s\right)^2 + \lambda\|h\|_{\mathcal{H}_{k_\mathcal{X}}}^2.$$

*On the other hand, solving* (9) *for $b = +\infty$ is equivalent to solve a unique kernel Ridge regression for all the tasks based on the entire dataset, i.e., we have*

$$\forall\, i \in [N], \qquad \mu_t(i, \cdot \mid b = +\infty) = \arg\min_{h \in \mathcal{H}_{k_\mathcal{X}}} \sum_{s=1}^t \left(h(x_s) - y_s\right)^2 + \lambda N\|h\|_{\mathcal{H}_{k_\mathcal{X}}}^2.$$

Hence, choosing some $b$ in $(0, +\infty)$ is choosing some tradeoff between these two extreme regimes. In Section 3.1, see Theorem 2, we discuss how to set $b$ with respect to the tasks at hand. Proposition 2 provides another intuition: to ensure constant regularization, parameter $\lambda$ should decrease with $b$. This key observation was overlooked in [6], partly explaining our better guarantees, see Section 3.1.

Another way to gain intuition about the setting is to compute the feature map induced by the multitask kernel $k$. Let $\phi \colon \mathcal{X} \to \mathbb{R}^d$ be the canonical feature map associated to $k_\mathcal{X}$[3], and $\psi \colon \mathcal{X} \times [N] \to \mathbb{R}^{Nd}$ such that $\psi(i, x) = (0, \ldots, \phi(x), \ldots, 0)$, with non-zero entry at block $i$. Further define $A_\otimes = A \otimes I_d \in \mathbb{R}^{Nd \times Nd}$, where $\otimes$ denote the kronecker product. It is immediate to check that

$$\widetilde{\psi}(i, x) = A_\otimes^{-1/2}\,\psi(i, x) = \left(A_{1i}^{-1/2}\,\phi(x), \ldots, A_{ii}^{-1/2}\,\phi(x), \ldots, A_{Ni}^{-1/2}\,\phi(x)\right) \in \mathbb{R}^{Nd} \tag{10}$$

is a feature map associated to $k$. Indeed, for any tasks $i, i' \in [N]$, and points $x, x' \in \mathcal{X}$, we have that $\langle \widetilde{\psi}(i, x), \widetilde{\psi}(i', x') \rangle = \sum_k A_{ik}^{-1/2} A_{i'k}^{-1/2} \langle \phi(x), \phi(x') \rangle = A_{ii'}^{-1} k_\mathcal{X}(x, x') = k\big((i, x), (i', x')\big)$. Hence, $\widetilde{\psi}(i, x)$ stores the feature map $\phi(x)$ in each block $j$, weighted by some coefficient $A_{ji}^{-1/2}(b)$,

---

[3]For the sake of the exposition we consider finite dimensional feature maps. Our results naturally extend to infinite dimensional ones.

which quantifies how similar tasks $i$ and $j$ are assumed to be. When $b = 0$, only block $i$ receives $\phi(x)$. When $b = +\infty$, each block receives $\phi(x)/N$. Denoting abusively $f = (f_1, \ldots, f_N)$, note that

$$f^{\text{mt}}(i, x) = \langle f_i, \phi(x) \rangle = \langle f, \psi(i, x) \rangle = \langle A_\otimes^{1/2} f, A_\otimes^{-1/2} \psi(i, x) \rangle = \langle \widetilde{f}, \widetilde{\psi}(i, x) \rangle, \qquad (11)$$

where $\widetilde{f} = \left( \sum_j A_{1j}^{1/2} f_j, \ldots, \sum_j A_{Nj}^{1/2} f_j \right) \in \mathcal{H}_k$ is a transformed representation of $f^{\text{mt}}$ that will play a key role in the subsequent analysis.

**Comparison to the model of [9].** We note that an online multitask kernel regression setting has also been considered in [9]. However, the setting of [9] considers *multioutput* functions $f^{\text{mo}} \colon \mathcal{X} \to \mathbb{R}^N$ such that the $i^{\text{th}}$ output of $f_{\text{mo}}$ is given by $f_i$. The authors learn $f^{\text{mo}}$ by leveraging matrix-valued kernels, a multiple output extension of scalar-valued kernel methods. Although the functions modeled by kernel $k$ defined in (1) are isomorphic to that associated to the matrix-valued decomposable kernel given by $k_{\text{MV}}(x, x') = k_{\mathcal{X}}(x, x') A^{-1}$, where we recall that $A_{ii'}^{-1} = k_{\mathcal{T}}(i, i')$, we highlight that both models are drastically different. Indeed, while we only observe a single measurement at each time step $t$, namely a noisy version of $f_{i_t}(x_t)$, the model in [9] assumes that the learner can access $N$ different measurements, that of the $f_i(x_t)$ for all $i \in [N]$. This key difference makes our model more flexible. In particular, in [9] all tasks must be observed the same number of times, and in addition at the same observation points.

### A.1.1 Proof of Proposition 2

Recall first that problem (9) writes

$$\mu_t = \arg\min_{h \in \mathcal{H}_k} \sum_{s=1}^t \big( h(i_s, x_s) - y_s \big)^2 + \lambda \|h\|_{\mathcal{H}_k}^2.$$

Using identity (11), we obtain

$$\mu_t = \arg\min_{h \in \mathcal{H}_k} \sum_{s=1}^t \Big( \langle h, \widetilde{\psi}(i_s, x_s) \rangle - y_s \Big)^2 + \lambda \|h\|_{\mathcal{H}_k}^2,$$

such that standard results for kernel Ridge regression gives that $\mu_t(x) = \langle w_t, \widetilde{\psi}(x) \rangle$, with

$$w_t = \left( \sum_{s=1}^t \widetilde{\psi}(i_s, x_s) \widetilde{\psi}(i_s, x_s)^\top + \lambda I_{Nd} \right)^{-1} \left( \sum_{s=1}^t y_s \widetilde{\psi}(i_s, x_s) \right). \qquad (12)$$

Now, recall that $\widetilde{\psi}(i, x) = A_\otimes^{-1/2} \psi(i, x) = \left( A_{1i}^{-1/2} \phi(x), \ldots, A_{ii}^{-1/2} \phi(x), \ldots, A_{Ni}^{-1/2} \phi(x) \right)$, such that

$$\widetilde{\psi}(i, x \,|\, b = 0) = (0, \ldots, 0, \underset{\text{block } i}{\phi(x)}, 0, \ldots, 0), \qquad (13)$$

$$\widetilde{\psi}(i, x \,|\, b = +\infty) = (\phi(x)/N, \ldots, \phi(x)/N). \qquad (14)$$

Substituting (13) in (12), we obtain

$$w_t = \begin{pmatrix} \sum_{\substack{s \leq t \\ \text{s.t. } i_s = 1}} \phi(x_s)\phi(x_s)^\top + \lambda I_d & & (0) \\ & \ddots & \\ (0) & & \sum_{\substack{s \leq t \\ \text{s.t. } i_s = N}} \phi(x_s)\phi(x_s)^\top + \lambda I_d \end{pmatrix}^{-1} \begin{pmatrix} \sum_{\substack{s \leq t \\ \text{s.t. } i_s = 1}} y_s \phi(x_s) \\ \vdots \\ \sum_{\substack{s \leq t \\ \text{s.t. } i_s = N}} y_s \phi(x_s) \end{pmatrix},$$

or again,

$$\forall i \in [N], \qquad w_t^{(i)} = \left( \sum_{s \leq t \,:\, i_s = i} \phi(x_s)\phi(x_s)^\top + \lambda I_d \right)^{-1} \left( \sum_{s \leq t \,:\, i_s = i} y_s \phi(x_s) \right),$$

where $w^{(i)} \in \mathbb{R}^d$ denotes the $i^{\text{th}}$ block of concatenated vector $w \in \mathbb{R}^{Nd}$. We thus recover the solutions to the independent regressions stated in the first claim of Proposition 2.

Alternatively, substituting (14) into (12), we obtain

$$
w_t = \begin{pmatrix} \frac{1}{N}\sum_{s=1}^{t}\phi(x_s)\phi(x_s)^\top + \lambda N\, I_d & & \frac{1}{N}\sum_{s=1}^{t}\phi(x_s)\phi(x_s)^\top \\ & \ddots & \\ \frac{1}{N}\sum_{s=1}^{t}\phi(x_s)\phi(x_s)^\top & & \frac{1}{N}\sum_{s=1}^{t}\phi(x_s)\phi(x_s)^\top + \lambda N\, I_d \end{pmatrix}^{-1} \begin{pmatrix} \sum_{s=1}^{t} y_s\phi(x_s) \\ \vdots \\ \sum_{s=1}^{t} y_s\phi(x_s) \end{pmatrix}
$$

$$
= \begin{pmatrix} \sum_{s=1}^{t}\phi(x_s)\phi(x_s)^\top + \lambda N\, I_d & & (0) \\ & \ddots & \\ (0) & & \sum_{s=1}^{t}\phi(x_s)\phi(x_s)^\top + \lambda N\, I_d \end{pmatrix}^{-1} \begin{pmatrix} \sum_{s=1}^{t} y_s\phi(x_s) \\ \vdots \\ \sum_{s=1}^{t} y_s\phi(x_s) \end{pmatrix},
$$

such that for any $i \in [N]$ we have $w_t^{(i)} = \left(\sum_{s=1}^{t}\phi(x_s)\phi(x_s)^\top + \lambda N\, I_d\right)^{-1}\left(\sum_{s=1}^{t} y_s\phi(x_s)\right)$, which are exactly the solutions to the regression problem on the full dataset stated in the second claim of Proposition 2. Note that the above equality can be easily checked by left multiplying both expressions by $\mathbb{1}_N\mathbb{1}_N^\top \otimes \frac{1}{N}\sum_{s=1}^{t}\phi(x_s)\phi(x_s)^\top + \lambda N\, I_{Nd}$. □

## A.2 Proof of Theorem 1

**Theorem 1** (Multitask confidence intervals). *Let $f^{\text{mt}} \colon [N] \times \mathcal{X} \to \mathbb{R}$ such that for all $i \in [N]$, $f_i := f^{\text{mt}}(i, \cdot)$ belongs to the RKHS associated to $k_{\mathcal{X}}$ and $\|f_i\| \leq B$. Moreover, let $\mu_t$ and $\sigma_t$ be the regression estimates of Equations (3) and (4) with task kernel $k_{\mathcal{T}}(i,j) = [K_{\text{task}}(b)]_{ij}$, parameter $\lambda \in [1/(1+b), 1]$, and noise $\{\xi_\tau\}_{\tau=1}^t$ i.i.d. 1-sub-Gaussian. Then, with probability at least $1 - 2\delta$,*

$$
\left|\mu_t(i, x \mid b) - f^{\text{mt}}(i, x)\right| \leq \beta_t^{\text{new}}(b) \cdot \sigma_t(i, x \mid b), \qquad \forall t \in \mathbb{N}, i \in [N], x \in \mathcal{X},
$$

*where* $\quad \beta_t^{\text{new}}(b) = \min\left\{\beta_t^{\text{naive}}(b), \beta_t^{\text{small-}b}(b), \beta_t^{\text{large-}b}(b)\right\},$

$$
\beta_t^{\text{small-}b}(b) = B(1 + b\epsilon)\sqrt{\frac{1 + bN}{1 + b}} + \lambda^{-1/2}\sqrt{2(1 + bN)\left(\gamma_t^{\text{st}} + \ln(N/\delta)\right)},
$$

$$
\beta_t^{\text{large-}b}(b) = B\sqrt{\frac{(1 + b\epsilon)^2}{1 + b} + \frac{2bN}{1 + b} + \frac{2b(1 + b\epsilon)^2}{N\lambda^2(1 + b)^3}\, t^2} + \lambda^{-1/2}\sqrt{2\left(\gamma_t^{\text{mt}}(b) + \ln(1/\delta)\right)}.
$$

*Proof.* Recall that we are interested in obtaining high probability error bounds of the form

$$
\left|\mu_t(i, x) - f^{\text{mt}}(i, x)\right| \leq \beta_t(b) \cdot \sigma_t(i, x \mid b)
$$

for a suitable choice of confidence width $\beta_t(b)$.

### A.2.1 A naive confidence width

Using identity (11) and a direct application of [1, Theorem 3.11 and Remark 3.13], together with the definition of $\gamma_t^{\text{mt}}(b)$, we can select

$$
\beta_t = \left\|\widetilde{f}\right\| + \lambda^{-1/2}\sqrt{2\left(\gamma_t^{\text{mt}}(b) + \ln(1/\delta)\right)}.
$$

We now upper bound $\left\|\widetilde{f}\right\|$ explicitly. Recall that

$$
\widetilde{f} = \left(\sum_j A_{1j}^{1/2} f_j, \ldots, \sum_j A_{Nj}^{1/2} f_j\right),
$$

with $A = (1 + b)I_N - (b/N)\mathbb{1}\mathbb{1}^\top$, so that we have

$$\big\|\widetilde{f}\big\|^2 = \sum_{i=1}^N \Big\| \sum_{j=1}^N A_{ij}^{1/2} f_j \Big\|^2 = \sum_{i=1}^N \sum_{j=1}^N \sum_{k=1}^N A_{ij}^{1/2} A_{ik}^{1/2} \langle f_j, f_k \rangle$$

$$= \sum_{j,k=1}^N A_{jk} \langle f_j, f_j \rangle = \sum_{i=1}^N \|f_i\|^2 + b \left( \sum_{i=1}^N \|f_i\|^2 - \frac{1}{N} \sum_{j,k=1}^N \langle f_j, f_k \rangle \right). \tag{15}$$

On the other side, it holds

$$\sum_{i=1}^N \Big\| f_i - \frac{1}{N} \sum_{j=1}^N f_j \Big\|^2 = \sum_{i=1}^N \left( \|f_i\|^2 + \frac{1}{N^2} \Big\| \sum_{j=1}^N f_j \Big\|^2 - \frac{2}{N} \sum_{j=1}^N \langle f_i, f_j \rangle \right)$$

$$= \sum_{i=1}^N \|f_i\|^2 + \frac{1}{N} \sum_{i,j=1}^N \langle f_i, f_j \rangle - \frac{2}{N} \sum_{i,j=1}^N \langle f_i, f_j \rangle$$

$$= \sum_{i=1}^N \|f_i\|^2 - \frac{1}{N} \sum_{i,j=1}^N \langle f_i, f_j \rangle. \tag{16}$$

Substituting (16) into (15), we obtain

$$\big\|\widetilde{f}\big\| = \sum_{i=1}^N \|f_i\|^2 + b \sum_{i=1}^N \|f_i - f_{\text{avg}}\|^2 \leq NB^2(1 + b\epsilon^2).$$

Overall, we can thus choose

$$\beta_t^{\text{naive}}(b) = B\sqrt{N(1 + b\epsilon^2)} + \lambda^{-1/2}\sqrt{2\big(\gamma_t^{\text{mt}}(b) + \ln(1/\delta)\big)}. \tag{17}$$

Note that this choice corresponds also to the confidence intervals utilized in the multitask regression setting of [6]. However, we highlight that width (17) is *too conservative*. Indeed, when setting $b = 0$ our model consists in solving single tasks independently, see Proposition 2, such that the confidence width should be of the order $B + \mathcal{O}(\sqrt{\gamma_t^{\text{st}}})$. Instead, we have

$$\beta_t^{\text{naive}}(0) = \sqrt{N}B + \mathcal{O}\big(\sqrt{\gamma_t^{\text{mt}}(0)}\big) = \sqrt{N}\left( B + \mathcal{O}\big(\sqrt{\gamma_t^{\text{st}}}\big) \right),$$

which is $\sqrt{N}$ bigger. In the next subsection, we derive an improved confidence width which is tight at $b = 0$.

### A.2.2 An improved confidence width

As in [8, 2, Proof of Theorem 2], we start by bounding the prediction error as

$$\big| f^{\text{mt}}(i, x) - \mu_t(i, x) \big| = \big| f^{\text{mt}}(i, x) - \boldsymbol{k}_t(i, x)^\top (K_t + \lambda I_t)^{-1}(\bar{\boldsymbol{y}}_{1:t} + \boldsymbol{\xi}_{1:t}) \big|$$

$$\leq \underbrace{\big| f^{\text{mt}}(i, x) - \boldsymbol{k}_t(i, x)^\top (K_t + \lambda I_t)^{-1} \bar{\boldsymbol{y}}_{1:t} \big|}_{\text{bias error}} + \underbrace{\big| \boldsymbol{k}_t(i, x)^\top (K_t + \lambda I_t)^{-1} \boldsymbol{\xi}_{1:t} \big|}_{\text{variance error}},$$

where $\bar{\boldsymbol{y}}_{1:t} \in \mathbb{R}^t$ is the vector of noise-free outputs such that $\bar{y}_s = f^{\text{mt}}(i_s, x_s) = f_{i_s}(x_s)$, and $\boldsymbol{\xi}_{1:t} \in \mathbb{R}^t$ is the vector of noises. Below, we derive separate bounds for the bias and variance errors. But before doing so, we depart from [8, Proof of Theorem 2] and obtain an explicit lower bound for the predictive variance for task $i$ at point $x$.

**Lower bounding the predictive variance.** First, we introduce some notation. Let $A_\otimes \in \mathbb{R}^{Nd \times Nd}$ such that

$$A_\otimes := A \otimes I_d = (1 + b)I_{Nd} - \frac{b}{N}\mathbb{1}_N \mathbb{1}_N^\top \otimes I_d,$$

Moreover, for every $t \in \mathbb{N}$ and $i \in [N]$, let $t_i = \sum_{s \leq t} \mathbb{I}\{i_s = i\}$. We denote by $\Phi_{t_i} \in \mathbb{R}^{t_i \times d}$ the matrix storing in rows the $\phi(x_s)$ for all $s$ such that $i_s = i$. Similarly, let $\Psi_t, \widetilde{\Psi}_t \in \mathbb{R}^{t \times Nd}$ storing in

rows the $\psi(i_s, x_s)$ and $\widetilde{\psi}(i_s, x_s)$ respectively, for all $s \leq t$. Recall that $\widetilde{\psi}(i, x) = A_\otimes \psi(i, x)$, such that $\widetilde{\Psi}_t = \Psi_t A_\otimes$. Further, define

$$M := \Psi_t^\top \Psi_t + \lambda(1+b)I_{Nd} = \begin{bmatrix} M_1 & & 0 \\ & \ddots & \\ 0 & & M_N \end{bmatrix}, \quad \text{with} \quad M_i = \Phi_{t_i}^\top \Phi_{t_i} + \lambda(1+b)I_d. \quad (18)$$

By properties of the multitask kernel, see (1), the predictive variance for task $i$ at decision point $x$ can be lower bounded as:

$$\sigma_t^2(i, x) = k\big((i, x), (i, x)\big) - \boldsymbol{k}_t(i, x)^\top (K_t + \lambda I_t)^{-1} \boldsymbol{k}_t(i, x)$$

$$= \widetilde{\psi}(i, x)^\top \left( I_{Nd} - \widetilde{\Psi}_t^\top \big( \widetilde{\Psi}_t \widetilde{\Psi}_t^\top + \lambda I_t \big)^{-1} \widetilde{\Psi}_t \right) \widetilde{\psi}(i, x)$$

$$= \widetilde{\psi}(i, x)^\top \left( I_{Nd} - \big( \widetilde{\Psi}_t^\top \widetilde{\Psi}_t + \lambda I_{Nd} \big)^{-1} \widetilde{\Psi}_t^\top \widetilde{\Psi}_t \right) \widetilde{\psi}(i, x) \quad (19)$$

$$= \lambda \widetilde{\psi}(i, x)^\top \big( \widetilde{\Psi}_t^\top \widetilde{\Psi}_t + \lambda I_{Nd} \big)^{-1} \widetilde{\psi}(i, x)$$

$$= \lambda \psi(i, x)^\top A_\otimes^{-1/2} \left( A_\otimes^{-1/2} \Psi_t^\top \Psi_t A_\otimes^{-1/2} + \lambda A_\otimes^{-1/2} A_\otimes A_\otimes^{-1/2} \right)^{-1} A_\otimes^{-1/2} \psi(i, x)$$

$$= \lambda \psi(i, x)^\top \big( \Psi_t^\top \Psi_t + \lambda A_\otimes \big)^{-1} \psi(i, x)$$

$$= \lambda \psi(i, x)^\top \left( M - \frac{\lambda b}{N} \mathbb{1}_N \mathbb{1}_N^\top \otimes I_d \right)^{-1} \psi(i, x)$$

$$= \lambda \psi(i, x)^\top M^{-1} \psi(i, x)$$

$$\quad + \lambda \psi(i, x)^\top M^{-1} \left( \mathbb{1}_N \mathbb{1}_N^\top \otimes \frac{\lambda b}{N} \left( I_{Nd} - \frac{\lambda b}{N} \sum_{i=1}^N M_i^{-1} \right)^{-1} \right) M^{-1} \psi(i, x) \quad (20)$$

$$= \lambda \phi(x)^\top M_i^{-1} \phi(x) + \lambda \phi(x)^\top M_i^{-1} \frac{\lambda b}{N} \underbrace{\left( I_d - \frac{\lambda b}{N} \sum_{i=1}^N M_i^{-1} \right)^{-1}}_{:=X^{-1}} M_i^{-1} \phi(x)$$

$$= \lambda \left( \big\| M_i^{-1/2} \phi(x) \big\|^2 + \frac{\lambda b}{N} \big\| X^{-1/2} M_i^{-1} \phi(x) \big\|^2 \right)$$

$$\geq \lambda \big\| M_i^{-1/2} \phi(x) \big\|^2 + \frac{b}{(1+b)N} \big\| \lambda X^{-1} M_i^{-1} \phi(x) \big\|^2 \quad (21)$$

$$\geq (1+b) \big\| \lambda M_i^{-1} \phi(x) \big\|^2 + \frac{b}{(1+b)N} \big\| \lambda X^{-1} M_i^{-1} \phi(x) \big\|^2, \quad (22)$$

where (19) comes from the *push-through* equality, (20) from Lemma 2 applied to $D = M$ and $P = -\frac{\lambda b}{N} I_d$, (21) from the fact that $X \succeq 1/(1+b) \, I_d$, and (22) from $M_i \succeq \lambda(1+b)I_d$. Indeed, the later can easily be checked from (18), which also implies that $(\lambda b)/N \sum_i M_i^{-1} \preceq b/(1+b) \, I_d$, such that $X \succeq 1/(1+b) \, I_d$. We now upper bound the bias error in terms of $\sigma_t^2(i, x)$.

**Bounding the bias error.** Using similar steps as before, we have

$$\big| f^{\mathrm{mt}}(i, x) - \boldsymbol{k}_t(i, x)^\top (K_t + \lambda I_t)^{-1} \bar{\boldsymbol{y}}_{1:t} \big|$$

$$= \big| \widetilde{\psi}(i, x)^\top \widetilde{f} - \widetilde{\psi}(i, x)^\top \widetilde{\Psi}_t^\top \big( \widetilde{\Psi}_t \widetilde{\Psi}_t^\top + \lambda I_t \big)^{-1} \widetilde{\Psi}_t \widetilde{f} \big|$$

$$= \big| \widetilde{\psi}(i, x)^\top \widetilde{f} - \widetilde{\psi}(i, x)^\top \big( \widetilde{\Psi}_t^\top \widetilde{\Psi}_t + \lambda I_{Nd} \big)^{-1} \widetilde{\Psi}_t^\top \widetilde{\Psi}_t \widetilde{f} \big|$$

$$= \big| \lambda \widetilde{\psi}(i, x)^\top \big( \widetilde{\Psi}_t^\top \widetilde{\Psi}_t + \lambda I_{Nd} \big)^{-1} \widetilde{f} \big|$$

$$= \big| \lambda \psi(i, x)^\top \big( \Psi_t^\top \Psi_t + \lambda A_\otimes \big)^{-1} A_\otimes^{1/2} \widetilde{f} \big|$$

$$= \left| \lambda \psi(i, x)^\top \left[ M^{-1} + M^{-1} \left( \mathbb{1}\mathbb{1}^\top \otimes \frac{\lambda b}{N} \left( I_d - \sum_{i=1}^N M_i^{-1} \right)^{-1} \right) M^{-1} \right] A_\otimes^{1/2} \widetilde{f} \right|$$

$$= \left| \left( \lambda M_i^{-1} \phi(x) \right)^\top \left[ A_\otimes^{1/2} \widetilde{f} \right]_{[i]} + \frac{\lambda b}{N} \left( \lambda X^{-1} M_i^{-1} \phi(x) \right)^\top \sum_{l=1}^N \left[ M^{-1} A_\otimes^{1/2} \widetilde{f} \right]_{[l]} \right|$$

$$\leq \left\| \lambda M_i^{-1} \phi(x) \right\| \cdot \left\| f_i + b(f_i - f_{\text{avg}}) \right\| + \frac{\lambda b}{N} \left\| \lambda X^{-1} M_i^{-1} \phi(x) \right\| \cdot \left\| \sum_{l=1}^N M_l^{-1} \left[ A_\otimes^{1/2} \widetilde{f} \right]_{[l]} \right\|$$

$$\leq \sqrt{1+b} \left\| \lambda M_i^{-1} \phi(x) \right\| \frac{B(1+b\epsilon)}{\sqrt{1+b}}$$

$$+ \sqrt{\frac{b}{(1+b)N}} \left\| \lambda X^{-1} M_i^{-1} \phi(x) \right\| \cdot \lambda \sqrt{\frac{b(1+b)}{N}} \underbrace{\left\| \sum_{l=1}^N M_l^{-1} \left[ A_\otimes^{1/2} \widetilde{f} \right]_{[l]} \right\|}_{:= \|D\|}$$

$$\leq \sqrt{ \left( (1+b) \left\| \lambda M_i^{-1} \phi(x) \right\|^2 + \frac{b \left\| \lambda X^{-1} M_i^{-1} \phi(x) \right\|^2}{(1+b)N} \right) \left( \frac{B^2(1+b\epsilon)^2}{1+b} + \frac{\lambda^2 b(1+b)}{N} \|D\|^2 \right) } \tag{23}$$

$$\leq \sigma_t(i,x) \cdot \sqrt{ \frac{B^2(1+b\epsilon)^2}{1+b} + \frac{\lambda^2 b(1+b)}{N} \|D\|^2 } , \tag{24}$$

where we have used $v_{[i]}$ to denote the block $i$ of a concatenated vector in $\mathbb{R}^{Nd}$, Cauchy-Schwarz inequality to derive (23), and lower bound (22) to obtain (24). Note that $D$ depends on the products between data matrices $M_l$ and task vectors $f_i, \ldots, f_N$, such that it is unknown in general. Below, we provide two upper bounds for $\|D\|$.

*Bound 1 (small $b$ range).* We can bound $\|D\|$ using Cauchy-Schwarz. We have

$$\|D\| \leq \sum_{l=1}^N \left\| M_l^{-1} \right\|_* \left\| \left[ A_\otimes^{1/2} \widetilde{f} \right]_{[l]} \right\| \leq \frac{1}{\lambda(1+b)} \sum_{l=1}^N \left\| f_l + b(f_l - f_{\text{avg}}) \right\| \leq \frac{NB(1+b\epsilon)}{\lambda(1+b)} ,$$

which yields

$$\left| f^{\text{mt}}(i,x) - k_t(i,x)^\top (K_t + \lambda I_t)^{-1} \bar{y}_{1:t} \right| \leq \sigma_t(i,x) \cdot B(1+b\epsilon) \sqrt{\frac{1+bN}{1+b}} . \tag{25}$$

The above bound is useful when $b$ is small. Indeed, when $b = 0$, according to the above we recover the single-task confidence width $B \cdot \sigma_t(i,x)$. However, as $b$ goes to $+\infty$, the bound grows as $\mathcal{O}(\sqrt{N}b\epsilon)$, which is an order of $\sqrt{b}$ faster than the naive one, see (17). We thus provide another upper bound on $\|D\|$, which is tighter for large values of $b$.

*Bound 2 (large $b$ range).* Alternatively, we can bound $\|D\|$ leveraging the SVD of the data matrices used to build $M_l$ (recall that $M_l = \lambda(1+b)I_d + \Phi_{t_l}^\top \Phi_{t_l}$). For $l \leq N$, let $\Phi_{t_l}^\top \Phi_{t_l} = \sum_k \sigma_k^{(l)} u_k^{(l)} u_k^{(l)\top}$ be the SVD of the data matrix $\Phi_{t_l}^\top \Phi_{t_l}$. We have

$$M_l^{-1} = \sum_k \frac{1}{\sigma_k^{(l)} + \lambda(1+b)} u_k^{(l)} u_k^{(l)\top} = \frac{1}{\lambda(1+b)} I_d - \sum_k \frac{\sigma_k^{(l)}}{\lambda(1+b)\left(\sigma_k^{(l)} + \lambda(1+b)\right)} u_k^{(l)} u_k^{(l)\top} ,$$

so that

$$\|D\| = \left\| \sum_{l=1}^N M_l^{-1} \left( f_l + b(f_l - f_{\text{avg}}) \right) \right\|$$

$$\leq \frac{1}{\lambda(1+b)} \left\| \sum_{l=1}^N f_l \right\| + \frac{1}{\lambda(1+b)} \left\| \sum_{k,l} \frac{\sigma_k^{(l)}}{\sigma_k^{(l)} + \lambda(1+b)} u_k^{(l)} u_k^{(l)\top} \left( f_l + b(f_l - f_{\text{avg}}) \right) \right\|$$

$$\leq \frac{NB}{\lambda(1+b)} + \frac{1}{\lambda(1+b)} \sum_{k,l} \frac{\sigma_k^{(l)}}{\sigma_k^{(l)} + \lambda(1+b)} \left\| f_l + b(f_l - f_{\text{avg}}) \right\|$$

$$\leq \frac{NB}{\lambda(1+b)} + \frac{B(1+b\epsilon)}{\lambda(1+b)} \sum_{l=1}^{N} \text{Tr}\left(K_{t_l}\big(K_{t_l} + \lambda(1+b)I_{t_l}\big)^{-1}\right) \tag{26}$$

$$\leq \frac{NB}{\lambda(1+b)} + \frac{B(1+b\epsilon)}{\lambda(1+b)} \sum_{l=1}^{N} \text{Tr}(K_{t_l}) \cdot \lambda_{\max}\left(\big(K_{t_l} + \lambda(1+b)I_{t_l}\big)^{-1}\right)$$

$$\leq \frac{NB}{\lambda(1+b)} + \frac{B(1+b\epsilon)}{\lambda^2(1+b)^2} \sum_{l=1}^{N} t_l$$

$$\leq \frac{NB}{\lambda(1+b)} + \frac{B(1+b\epsilon)}{\lambda^2(1+b)^2}\, t\,.$$

We note that in practice the data-dependent bound (26) might be tighter than the latter one when $\lambda$ is small. However, for simplicity of the exposition we focus on the latter data-independent bound. Substituting it into (24), we obtain

$$\left| f^{\text{mt}}(i,x) - \boldsymbol{k}_t(i,x)^\top (K_t + \lambda I_t)^{-1} \bar{\boldsymbol{y}}_{1:t} \right|$$

$$\leq \sigma_t(i,x) \cdot \sqrt{\frac{B^2(1+b\epsilon)^2}{1+b} + 2B^2 \frac{bN}{1+b} + 2t^2 \frac{B^2(1+b\epsilon)^2 b}{\lambda^2(1+b)^3 N}}$$

$$= \sigma_t(i,x) \cdot B\sqrt{\frac{(1+b\epsilon)^2}{1+b} + \frac{2bN}{1+b} + \frac{2b(1+b\epsilon)^2}{N\lambda^2(1+b)^3}\, t^2}\,. \tag{27}$$

When $b$ goes to 0, we recover $B \cdot \sigma_t(i,x)$, as for Bound 1. However, the above bound is more useful when $b$ is large. Indeed, when $b$ goes to $+\infty$, we obtain $B\sqrt{b\epsilon^2 + 2N + 2\epsilon^2 t^2/N\lambda^2}\, \sigma_t(i,x) = \mathcal{O}(B\sqrt{b}\epsilon)\, \sigma_t(i,x)$, which improves by a factor $\sqrt{N}$ over the $\mathcal{O}(B\sqrt{bN}\epsilon)\, \sigma_t(i,x)$ term obtained with the naive bound. Recall that when $b$ goes to $+\infty$, MT regression is equivalent to solve a single averaged task based on the whole dataset, see Proposition 2, such that obtaining a confidence width independent from $N$ is expected. Overall, combining (17), (25) and (27), we obtain that

$$\left| f^{\text{mt}}(i,x) - \boldsymbol{k}_t(i,x)^\top (K_t + \lambda I_t)^{-1} \bar{\boldsymbol{y}}_{1:t} \right| \leq \beta_t^{\text{bias}}(b) \cdot \sigma_t(i,x)\,, \tag{28}$$

where $\beta_t^{\text{bias}}(b) = B \min\left\{ \sqrt{N(1+b\epsilon^2)},\ (1+b\epsilon)\sqrt{\frac{1+bN}{1+b}},\ \sqrt{\frac{(1+b\epsilon)^2}{1+b} + \frac{2bN}{1+b} + \frac{2b(1+b\epsilon)^2}{N\lambda^2(1+b)^3}\, t^2} \right\}$. We now turn to the variance error.

**Bounding the variance error.** Using (17), the variance error can be naively bounded by $\beta_t^{\text{naive, var}}(b) \cdot \sigma_t(i,x)$, where $\beta_t^{\text{naive, var}}(b) = \lambda^{-1/2}\sqrt{2\big(\gamma_t^{\text{mt}}(b) + \ln(1/\delta)\big)}$. However, note that the above bound is conservative when parameter $b$ is small. Indeed, in the limit of $b = 0$ (tasks are treated independently), we know that such error should only depend on the information gain of task $i$, i.e., $\gamma_{t_i}^{\text{st}}$. Instead, $\gamma_t^{\text{mt}}(0) = \mathcal{O}(N\gamma_{t_i}^{\text{st}})$, which is $N$ times bigger. Let $P_i := \Phi_{t_i}^\top \Phi_{t_i} + I_d$, and note that $M_i^{-1} \preceq P_i^{-1}$, since $\lambda \geq 1/(1+b)$. Looking at a single task $i$, with probability $1-\delta$ we have

$$\left\| M_i^{-1/2} \Phi_{t_i}^\top \boldsymbol{\xi}_{[t_i]} \right\|^2 \leq \left\| P_i^{-1/2} \Phi_{t_i}^\top \boldsymbol{\xi}_{[t_i]} \right\|^2$$

$$= \boldsymbol{\xi}_{[t_i]}^\top \Phi_{t_i} P_i^{-1} \Phi_{t_i}^\top \boldsymbol{\xi}_{[t_i]}$$

$$= \boldsymbol{\xi}_{[t_i]}^\top \Phi_{t_i} \big(\Phi_{t_i}^\top \Phi_{t_i} + I_d\big)^{-1} \Phi_{t_i}^\top \boldsymbol{\xi}_{[t_i]}$$

$$= \boldsymbol{\xi}_{[t_i]}^\top \big(K_{t_i} + I_{t_i}\big)^{-1} K_{t_i}\, \boldsymbol{\xi}_{[t_i]}$$

$$= \boldsymbol{\xi}_{[t_i]}^\top \big(I_{t_i} + K_{t_i}^{-1}\big)^{-1} \boldsymbol{\xi}_{[t_i]}$$

$$\leq 2\left(\frac{1}{2} \ln\left|I_{t_i} + K_{t_i}\right| + \ln(1/\delta)\right) \tag{29}$$

$$\leq 2\left(\frac{1}{2} \ln\left|I_{t_i} + \lambda^{-1}K_{t_i}\right| + \ln(1/\delta)\right) \tag{30}$$

$$\leq 2\big(\gamma_{t_i}^{\text{st}} + \ln(1/\delta)\big)\,,$$

where $\boldsymbol{\xi}_{[t_i]} \in \mathbb{R}^{t_i}$ contains the observation noises related to the time steps when task $i$ was active, and $K_{t_i} \in \mathbb{R}^{t_i \times t_i}$ is the individual Gram matrix based on such observations. Equation (29) is obtained by applying [8, Theorem 1] with $\eta = 0$, while (30) derives from $\lambda \leq 1$. By the union bound, we get that with probability at least $1 - \delta$, we have

$$\sup_{i \leq N} \left\| P_i^{-1/2} \Phi_{t_i}^\top \boldsymbol{\xi}_{[t_i]} \right\|^2 \leq 2\big(\gamma_t^{\mathrm{st}} + \ln(N/\delta)\big).$$

Then, we obtain

$$
\begin{aligned}
\Big| k_t&(i,x)^\top (K_t + \lambda I_t)^{-1} \boldsymbol{\xi}_{1:t} \Big| \\
&= \left| \widetilde{\psi}(i,x)^\top \big(\widetilde{\Psi}_t^\top \widetilde{\Psi}_t + \lambda I_{Nd}\big)^{-1} \widetilde{\Psi}_t^\top \boldsymbol{\xi}_{1:t} \right| \\
&= \left| \psi(i,x)^\top \big(\Psi_t^\top \Psi_t + \lambda A_\otimes\big)^{-1} \Psi_t^\top \boldsymbol{\xi}_{1:t} \right| \\
&= \left| \phi(x)^\top M_i^{-1} \Phi_{t_i}^\top \boldsymbol{\xi}_{[t_i]} + \frac{\lambda b}{N} \big(X^{-1} M_i^{-1} \phi(x)\big)^\top \sum_{l=1}^{N} \big[M^{-1} \Psi_t^\top \boldsymbol{\xi}_{1:t}\big]_{[l]} \right| \\
&\leq \sqrt{\lambda} \left\| M_i^{-1/2} \phi(x) \right\| \cdot \frac{1}{\sqrt{\lambda}} \left\| M_i^{-1/2} \Phi_{t_i}^\top \boldsymbol{\xi}_{[t_i]} \right\| \\
&\quad + \sqrt{\frac{b}{(1+b)N}} \left\| \lambda X^{-1} M_i^{-1} \phi(x) \right\| \cdot \sqrt{\frac{b(1+b)}{N}} \left\| \sum_{l=1}^{N} M_l^{-1} \Phi_{t_l}^\top \boldsymbol{\xi}_{[t_l]} \right\| \\
&\leq \sqrt{\lambda} \left\| M_i^{-1/2} \phi(x) \right\| \cdot \frac{1}{\sqrt{\lambda}} \left\| M_i^{-1/2} \Phi_{t_i}^\top \boldsymbol{\xi}_{[t_i]} \right\| \\
&\quad + \sqrt{\frac{b}{(1+b)N}} \left\| \lambda X^{-1} M_i^{-1} \phi(x) \right\| \cdot \sqrt{b(1+b)N} \sup_{l \leq N} \left\| M_l^{-1/2} \right\|_* \cdot \left\| M_l^{-1/2} \Phi_{t_l}^\top \boldsymbol{\xi}_{[t_l]} \right\| \\
&\leq \sqrt{\lambda} \left\| M_i^{-1/2} \phi(x) \right\| \cdot \frac{1}{\sqrt{\lambda}} \left\| M_i^{-1/2} \Phi_{t_i}^\top \boldsymbol{\xi}_{[t_i]} \right\| \\
&\quad + \sqrt{\frac{b}{(1+b)N}} \left\| \lambda X^{-1} M_i^{-1} \phi(x) \right\| \cdot \sqrt{\frac{bN}{\lambda}} \sup_{l \leq N} \left\| M_l^{-1/2} \Phi_{t_l}^\top \boldsymbol{\xi}_{[t_l]} \right\| \\
&\leq \left( \lambda \left\| M_i^{-1/2} \phi(x) \right\|^2 + \frac{b}{(1+b)N} \left\| \lambda X^{-1} M_i^{-1} \phi(x) \right\|^2 \right)^{1/2} \\
&\quad \cdot \left( \frac{1}{\lambda} \left\| P_i^{-1/2} \Phi_{t_i}^\top \boldsymbol{\xi}_{[t_i]} \right\|^2 + \frac{bN}{\lambda} \sup_{l \leq N} \left\| P_l^{-1/2} \Phi_{t_l}^\top \boldsymbol{\xi}_{[t_l]} \right\|^2 \right)^{1/2} \\
&\leq \lambda^{-1/2} \sigma_t(i,x) \sup_{l \leq N} \left\| P_l^{-1/2} \Phi_{t_l}^\top \boldsymbol{\xi}_{[t_l]} \right\| \sqrt{1 + bN} \qquad\qquad (31) \\
&\leq \lambda^{-1/2} \sqrt{2(1 + bN)\big(\gamma_t^{\mathrm{st}} + \ln(N/\delta)\big)} \, \sigma_t(i,x),
\end{aligned}
$$

where (31) comes from lower bound (21). Finally, we can take the minimum of this bound and the naive one. Using the union bound again, with probability at least $1 - 2\delta$, we have

$$\left| \boldsymbol{k}_t(i,x)^\top (K_t + \lambda I_t)^{-1} \boldsymbol{\xi}_{1:t} \right| \leq \beta_t^{\mathrm{var}}(b) \cdot \sigma_t(i,x), \qquad\qquad (32)$$

where $\beta_t^{\mathrm{var}}(b) = \lambda^{-1/2} \min\left\{ \sqrt{2\big(\gamma_t^{\mathrm{mt}}(b) + \ln(1/\delta)\big)}, \sqrt{2(1 + bN)\big(\gamma_t^{\mathrm{st}} + \ln(N/\delta)\big)} \right\}$.

**Overall error bound.** We can obtain the overall prediction error bound by combining bounds (28) and (32) for the bias and variance errors respectively. Hence, with probability $1 - 2\delta$, we have

$$
\begin{aligned}
\left| f^{\mathrm{mt}}(i,x) - \mu_t(i,x) \right| &= \left| f^{\mathrm{mt}}(i,x) - \boldsymbol{k}_t(i,x)^\top (K_t + \lambda I_t)^{-1} (\bar{\boldsymbol{y}}_{1:t} + \boldsymbol{\xi}_{1:t}) \right| \\
&\leq \underbrace{\left| f^{\mathrm{mt}}(i,x) - \boldsymbol{k}_t(i,x)^\top (K_t + \lambda I_t)^{-1} \bar{\boldsymbol{y}}_{1:t} \right|}_{\text{bias error}} + \underbrace{\left| \boldsymbol{k}_t(i,x)^\top (K_t + \lambda I_t)^{-1} \boldsymbol{\xi}_{1:t} \right|}_{\text{variance error}} \\
&\leq \big(\beta_t^{\mathrm{bias}}(b) + \beta_t^{\mathrm{var}}(b)\big) \cdot \sigma_t(i,x),
\end{aligned}
$$

with $\beta_t^{\text{bias}}(b) = B \min \left\{ \sqrt{N(1+b\epsilon^2)}, (1+b\epsilon)\sqrt{\frac{1+bN}{1+b}}, \sqrt{\frac{(1+b\epsilon)^2}{1+b} + \frac{2bN}{1+b} + \frac{2b(1+b\epsilon)^2}{N\lambda^2(1+b)^3} t^2} \right\}$, and $\beta_t^{\text{var}}(b) = \lambda^{-1/2} \min \left\{ \sqrt{2(\gamma_t^{\text{mt}}(b) + \ln(1/\delta))}, \sqrt{2(1+bN)(\gamma_t^{\text{st}} + \ln(N/\delta))} \right\}$. In particular, we obtain Theorem 1 by considering specific combinations among the minimums involved in the definitions of $\beta_t^{\text{bias}}(b)$ and $\beta_t^{\text{var}}(b)$. The resulting $\beta_t^{\text{small-}b}(b)$ and $\beta_t^{\text{large-}b}(b)$ are chosen to be tight when $b$ goes to 0 or $+\infty$ respectively. □

### A.2.3  A Kronecker Sherman-Morrison Lemma

We now provide a lemma which extends the Sherman-Morrison formula to kronecker matrices.

**Lemma 2.** *Let* $D_1, \ldots, D_N \in \mathbb{R}^{d \times d}$ *be invertible,* $D = \text{diag}(D_1, \ldots, D_N) \in \mathbb{R}^{Nd \times Nd}$, *and* $P \in \mathbb{R}^{d \times d}$ *that commutes with* $D_i$ *for all* $i \in [N]$. *Then we have*

$$\left( D + \mathbb{1}_N \mathbb{1}_N^\top \otimes P \right)^{-1} = D^{-1} + D^{-1} \left( \mathbb{1}_N \mathbb{1}_N^\top \otimes Q \right) D^{-1},$$

*where* $Q = -\left( I_d + P(D_1^{-1} + \ldots + D_N^{-1}) \right)^{-1} P = -P \left( I_d + P(D_1^{-1} + \ldots + D_N^{-1}) \right)^{-1}$.

*Proof.* It is immediate to check that $\left( D + \mathbb{1}_N \mathbb{1}_N^\top \otimes P \right) \left( D^{-1} + D^{-1} \left( \mathbb{1}_N \mathbb{1}_N^\top \otimes Q \right) D^{-1} \right) = \left( D^{-1} + D^{-1} \left( \mathbb{1}_N \mathbb{1}_N^\top \otimes Q \right) D^{-1} \right) \left( D + \mathbb{1}_N \mathbb{1}_N^\top \otimes P \right) = I_{Nd}$. □

## B  New Guarantees for Multitask Online Learning

**Details on the independent regret bound.** We analyze the regret of the strategy which runs $N$ independent instances of IGP-UCB [8], one per task. Choosing the single-task confidence width $\beta_t^{\text{st}} = B + \sqrt{2(\gamma_t^{\text{st}} + \ln(N/\delta))}$, one can control the individual task regrets and obtain with probability at least $1 - \delta$:

$$R^{\text{mt}}(T) = \sum_{i=1}^{N} \sum_{t: \, i_t = i} \max_{x \in \mathcal{X}} f^{\text{mt}}(i, x) - f^{\text{mt}}(i, x_t)$$

$$\leq 4 \sum_{i=1}^{N} \beta_{T_i}^{\text{st}} \sqrt{T_i \, \gamma_{T_i}^{\text{st}}}$$

$$\leq 6 \left( B \sqrt{NT\gamma_T^{\text{st}}} + \sqrt{NT\gamma_T^{\text{st}}(\gamma_T^{\text{st}} + \ln(N/\delta))} \right),$$

where we have used that $\gamma_{T_i}^{\text{st}} \leq \gamma_T^{\text{st}}$ and Jensen's inequality. We exactly recover the first bound in Theorem 2.

### B.1  Proof of Lemma 1

**Lemma 1.** *Suppose that* $\lambda \geq (N+b)/(N+bN)$, *and that for all tasks* $i$, *point* $x$, *and time* $t$, *we have* $f^{\text{mt}}(i, x \,|\, b) \in [\, \mu_t(i, x \,|\, b) \pm \beta_t(b) \cdot \sigma_t(i, x \,|\, b)\,]$. *Then, the multitask regret of* MT-UCB *satisfies*

$$R^{\text{mt}}(T) \leq 4 \, \beta_T(b) \sqrt{\lambda \, T \gamma_T^{\text{mt}}(b)}.$$

*Proof.* The proof follows from standard arguments, see e.g., [25, Theorem 1] and [8, Theorem 3], reproduced here for completeness. For, $i \in [N]$, let $x_i^* = \arg\max_{x \in \mathcal{X}} f^{\text{mt}}(i, x)$. With probability $1 - \delta$, we have

$$\sum_{t=1}^{T} \max_{x \in \mathcal{X}} f^{\text{mt}}(i_t, x) - f^{\text{mt}}(i_t, x_t)$$

$$\leq \sum_{t=1}^{T} \text{ucb}_{t-1}(i_t, x_{i_t}^* \,|\, b) - \text{ucb}_{t-1}(i_t, x_t \,|\, b) + 2\beta_t(b) \cdot \sigma_{t-1}(i_t, x_t \,|\, b)$$

$$\leq 2 \sum_{t=1}^{T} \beta_t(b) \cdot \sigma_{t-1}(i_t, x_t \,|\, b)$$

$$\leq 2\beta_T(b) \sqrt{T \sum_{t=1}^{T} \sigma_{t-1}^2(i_t, x_t \,|\, b)} \,. \tag{33}$$

The main twist is that our regularization parameter $\lambda$ might be smaller than 1, preventing from a direct adaptation of [8, Lemma 4] to bound $\sum_{t=1}^{T} \sigma_{t-1}^2(i_t, x_t \,|\, b)$. However, this happens not to be a problem since our multitask predictive variance are also smaller than in the single-task case. Indeed, as long as $\lambda \geq (N + b)/(N + bN) = A(b)_{ii}^{-1}$, we have for all $i$ and $x$

$$\sigma_t^2(i, x) = k\big((i, x), (i, x)\big) - \boldsymbol{k}_t(i, x)^\top (K_t + \lambda I_t)^{-1} \boldsymbol{k}_t(i, x)$$
$$\leq k\big((i, x), (i, x)\big)$$
$$= A_{ii}^{-1} \, k_{\mathcal{X}}(x, x)$$
$$\leq \lambda \,.$$

Therefore, we have

$$\sum_{t=1}^{T} \sigma_{t-1}^2(i_t, x_t) = \lambda \sum_{t=1}^{T} \lambda^{-1} \sigma_{t-1}^2(i_t, x_t) \leq 2\lambda \sum_{t=1}^{T} \ln\big(1 + \lambda^{-1} \sigma_{t-1}^2(i_t, x_t)\big) \leq 4\lambda \, \gamma_T^{\mathrm{mt}}(b) \,, \tag{34}$$

where we have used that $x \leq \ln(1 + x)$ for any $x \in [0, 1]$, applied to the $\lambda^{-1} \sigma_{t-1}^2(i_t, x_t) \leq 1$, and [8, Lemma 3]. Substituting (34) into (33) concludes the proof. $\qquad\square$

## B.2 Proof of Proposition 1

**Proposition 1.** *Let $\lambda \leq 1$, $N \geq 2$, and $T_i \geq 1$ for all $i \in [N]$. Then, for any $b \geq 0$, we have*

$$\gamma_T^{\mathrm{mt}}(b) \leq N\gamma_T^{\mathrm{st}} - \frac{Nb}{8(1 + b)}, \qquad \text{and} \qquad \gamma_T^{\mathrm{mt}}(b) \leq \gamma_T^{\mathrm{st}} + \frac{T}{2\lambda(1 + b)}\,.$$

*Proof.* Recall that the multitask kernel writes $k\big((i, x), (i', x')\big) = k_{\mathcal{T}}(i, i') \cdot k_{\mathcal{X}}(x, x')$. Hence, the multitask Gram matrix $K_T$ can be written as $K_T = K_{\mathcal{T}} \odot K_{\mathcal{X}}$, where $K_{\mathcal{T}}, K_{\mathcal{X}} \in \mathbb{R}^{T \times T}$ are the task (respectively domain) Gram matrices. Moreover, up to rearranging the rows and columns of $K_{\mathcal{T}}$ (which does not change the determinant), we can assume that points are ordered by task activations. Let $T_i = \sum_{t=1}^{T} \mathbb{I}\{i_t = i\}$ be the number of times task $i$ has been queried. We have

$$K_{\mathcal{T}} = \begin{pmatrix} \frac{b+N}{(1+b)N} \, \mathbb{1}_{T_1} \mathbb{1}_{T_1}^\top & & \frac{b}{(1+b)N} \, \mathbb{1}_{T_1} \mathbb{1}_{T-T_1}^\top \\ & \ddots & \\ \frac{b}{(1+b)N} \, \mathbb{1}_{T_N} \mathbb{1}_{T-T_N}^\top & & \frac{b+N}{(1+b)N} \, \mathbb{1}_{T_N} \mathbb{1}_{T_N}^\top \end{pmatrix}$$

$$= \frac{b}{1+b} \frac{\mathbb{1}_T \mathbb{1}_T^\top}{N} + \frac{1}{1+b} \begin{pmatrix} \mathbb{1}_{T_1} \mathbb{1}_{T_1}^\top & & 0 \\ & \ddots & \\ 0 & & \mathbb{1}_{T_N} \mathbb{1}_{T_N}^\top \end{pmatrix}\,.$$

We also introduce the block notation $K_{\mathcal{X}}^{(i,j)} \in \mathbb{R}^{T_i \times T_j}$ and $K_{\mathcal{X}}^{\mathrm{diag}} \in \mathbb{R}^{T \times T}$ such that

$$K_{\mathcal{X}} = \begin{pmatrix} & & \\ & K_{\mathcal{X}}^{(i,j)} & \\ & & \end{pmatrix}, \qquad \text{and} \qquad K_{\mathcal{X}}^{\mathrm{diag}} = \begin{pmatrix} K_{\mathcal{X}}^{(1,1)} & & 0 \\ & \ddots & \\ 0 & & K_{\mathcal{X}}^{(N,N)} \end{pmatrix}\,.$$

Our bounds are based on the observation that $g \colon t \mapsto \ln|tX + (1-t)Y|$ is concave, with derivative $g'(t) = \mathrm{Tr}\Big(\big(tX + (1-t)Y\big)^{-1}(X - Y)\Big)$. Applying the concavity inequality at $t = 0$ and $t = 1$, we obtain that for any positive semi-definite matrices $X$ and $Y$ and any $t \in [0, 1]$ we have

$$\ln|tX + (1-t)Y| \leq \ln|Y| + t \cdot \Big(\mathrm{Tr}\big(Y^{-1}X\big) - T\Big), \tag{35}$$

$$\ln|tX + (1-t)Y| \le \ln|X| + (1-t) \cdot \left(\text{Tr}(X^{-1}Y) - T\right). \tag{36}$$

Hence, for any $b \ge 0$ we have

$$
\begin{aligned}
2\gamma_T^{\text{mt}}(b) &= \ln\left|I_T + \lambda^{-1}K_T\right| \\
&= \ln\left|I_T + \lambda^{-1}K_{\mathcal{T}} \odot K_{\mathcal{X}}\right| \\
&= \ln\left|I_T + \lambda^{-1}\left(\frac{b}{1+b}\frac{K_{\mathcal{X}}}{N} + \frac{1}{1+b}K_{\mathcal{X}}^{\text{diag}}\right)\right| \\
&= \ln\left|\frac{b}{1+b}\left(I_T + \lambda^{-1}\frac{K_{\mathcal{X}}}{N}\right) + \frac{1}{1+b}\left(I_T + \lambda^{-1}K_{\mathcal{X}}^{\text{diag}}\right)\right| \\
&\le \ln\left|I_T + \lambda^{-1}K_{\mathcal{X}}^{\text{diag}}\right| + \frac{b}{1+b}\left[\text{Tr}\left(\left(I_T + \lambda^{-1}K_{\mathcal{X}}^{\text{diag}}\right)^{-1}\left(I_T + \lambda^{-1}\frac{K_{\mathcal{X}}}{N}\right)\right) - T\right] \tag{37} \\
&= 2\sum_{i=1}^{N}\gamma_{T_i}^{\text{st}} + \frac{b}{1+b}\left[\text{Tr}\left(\left(I_T + \lambda^{-1}K_{\mathcal{X}}^{\text{diag}}\right)^{-1}\left(I_T + \lambda^{-1}\frac{K_{\mathcal{X}}}{N}\right)\right) - T\right], \tag{38}
\end{aligned}
$$

where (37) derives from (35). We now take a closer look at the second term. We have

$$
\begin{aligned}
&\text{Tr}\left(\left(I_T + \lambda^{-1}K_{\mathcal{X}}^{\text{diag}}\right)^{-1}\left(I_T + \lambda^{-1}\frac{K_{\mathcal{X}}}{N}\right)\right) \\
&= \text{Tr}\left(\begin{pmatrix}\left(I_{T_1} + \lambda^{-1}K_{\mathcal{X}}^{(1,1)}\right)^{-1} & & 0 \\ & \ddots & \\ 0 & & \left(I_{T_N} + \lambda^{-1}K_{\mathcal{X}}^{(N,N)}\right)^{-1}\end{pmatrix}\left(I_T + \lambda^{-1}\frac{K_{\mathcal{X}}}{N}\right)\right) \\
&= \sum_{i=1}^{N}\text{Tr}\left(\left(I_{T_i} + \lambda^{-1}K_{\mathcal{X}}^{(i,i)}\right)^{-1}\left(I_{T_i} + \lambda^{-1}\frac{K_{\mathcal{X}}^{(i,i)}}{N}\right)\right) \\
&= \sum_{i=1}^{N}\sum_{\tau=1}^{T_i}\frac{\lambda + \sigma_\tau^{(i)}/N}{\lambda + \sigma_\tau^{(i)}} \tag{39}
\end{aligned}
$$

where $\left\{\sigma_\tau^{(i)}\right\}_{\tau \le T_i}$ are the eigenvalues of $K_{\mathcal{X}}^{(i,i)}$, possibly equal to 0. For any $i$, let $F^{(i)}\colon \mathbb{R}^{T_i} \to \mathbb{R}$ the functions which to any $\boldsymbol{\sigma} = (\sigma_1, \ldots, \sigma_{T_i})$ associates $F^{(i)}(\boldsymbol{\sigma}) = \sum_{\tau=1}^{T_i}(\lambda + \sigma_\tau/N)/(\lambda + \sigma_\tau)$. For any $\tau_0, \tau_1$, we have

$$
\begin{aligned}
&(\sigma_{\tau_0} - \sigma_{\tau_1})\left(\frac{\partial F^{(i)}(\boldsymbol{\sigma})}{\partial \sigma_{\tau_0}} - \frac{\partial F^{(i)}(\boldsymbol{\sigma})}{\partial \sigma_{\tau_1}}\right) \\
&= (\sigma_{\tau_0} - \sigma_{\tau_1})\left(\frac{(1/N)(\lambda + \sigma_{\tau_0}) - (\lambda + \sigma_{\tau_0}/N)}{(\lambda + \sigma_{\tau_0})^2} - \frac{(1/N)(\lambda + \sigma_{\tau_1}) - (\lambda + \sigma_{\tau_1}/N)}{(\lambda + \sigma_{\tau_1})^2}\right) \\
&= \lambda\frac{N-1}{N}(\sigma_{\tau_1} - \sigma_{\tau_0})\left(\frac{1}{(\lambda + \sigma_{\tau_0})^2} - \frac{1}{(\lambda + \sigma_{\tau_1})^2}\right) \\
&\ge 0,
\end{aligned}
$$

such that $F^{(i)}$ is Schur-convex. Hence, (39) is maximized at eigenvalues of the form $(T_i, 0, \ldots, 0)$, since the latter majorizes any other admissible distribution of the eigenvalues (recall that we must have $\sigma_\tau^{(i)} \ge 0$ and $\sum_{\tau=1}^{T_i}\sigma_\tau^{(i)} = \text{Tr}(K_{\mathcal{X}}^{(i,i)}) \le T_i$), with value

$$\sum_{i=1}^{N}(T_i - 1) + \frac{\lambda + T_i/N}{\lambda + T_i} \le T - N + N\frac{1 + 1/N}{2} \le T - \frac{N}{4},$$

where we have used that $\lambda \le 1$, $T_i \ge 1$, and $N \ge 2$. Substituting into (38), we finally obtain

$$\gamma_T^{\text{mt}}(b) \le N\gamma_T^{\text{st}} - \frac{Nb}{8(1+b)}. \tag{40}$$

The second bound is obtained by modifying (37). Instead, we now apply (36) and obtain

$$
2\gamma_T^{\mathrm{mt}}(b) \leq \ln\left|I_T + \lambda^{-1}\frac{K_\mathcal{X}}{N}\right| + \frac{1}{1+b}\left[\mathrm{Tr}\left(\left(I_T + \lambda^{-1}\frac{K_\mathcal{X}}{N}\right)^{-1}\left(I_T + \lambda^{-1}K_\mathcal{X}{}^{\mathrm{diag}}\right)\right) - T\right]
$$

$$
\leq 2\,\gamma_T^{\mathrm{st}} + \frac{1}{1+b}\left(\mathrm{Tr}\left(I_T + \lambda^{-1}K_\mathcal{X}^{\mathrm{diag}}\right) - T\right) \tag{41}
$$

$$
\leq 2\,\gamma_T^{\mathrm{st}} + \frac{T}{\lambda(1+b)}
$$

where (41) comes from von Neumann's trace inequality and is tight when $N \to +\infty$. $\qquad\square$

## B.3 Proof of Theorem 2

**Theorem 2.** *Assume that $B \geq 1$, and that* MT-UCB *is run with $\beta_t = \beta_t^{\mathrm{new}}$ from Theorem 1, and $\lambda = (N+b)/(N+bN)$. Let $b = N/\epsilon^2$ if $T \leq N$, $b = 1/\epsilon^2$ if $T \geq N$ and $\epsilon \leq N^{-1/4}T^{-1/2}$, and $b = 0$ otherwise. Let $R^{\mathrm{st}}(T) = B\sqrt{T\gamma_T^{\mathrm{st}}} + \sqrt{T\gamma_T^{\mathrm{st}}\big(\gamma_T^{\mathrm{st}} + \ln(1/\delta)\big)}$ be the single task regret bound achieved by* IGP-UCB *(up to constant factors). Then, there exists a universal constant $C$ such that with probability $1 - 2\delta$ we have (up to $\log N$ factors)*

$$
R^{\mathrm{mt}}(T) \leq C \min\left\{\sqrt{N}R^{\mathrm{st}}(T)\ ,\ R^{\mathrm{st}}(T) + \epsilon BT^{3/2}\Big(\sqrt{\gamma_T^{\mathrm{st}} + \ln(1/\delta)} + \epsilon\sqrt{T}\Big)\ ,\right.
$$
$$
\left. R^{\mathrm{st}}(T) + \epsilon BT\sqrt{N}\Big(\sqrt{\gamma_T^{\mathrm{st}} + \ln(1/\delta)} + \epsilon\sqrt{NT}\Big)\right\}.
$$

*Proof.* From Lemma 1 and the choice $\beta_t = \beta_t^{\mathrm{new}}$, we have with probability at least $1 - 2\delta$

$$
R^{\mathrm{mt}}(T) \leq 4\,\beta_T^{\mathrm{new}}(b)\sqrt{\lambda\,T\,\gamma_T^{\mathrm{mt}}(b)}\,.
$$

**First bound, recovering independent learning.** Hence, in particular, we have

$R^{\mathrm{mt}}(T)$

$$
\leq 4\,\beta_T^{\mathrm{small}\text{-}b}(b)\sqrt{\lambda\,T\,\gamma_T^{\mathrm{mt}}(b)}
$$
$$
= 4\left(B(1+b\epsilon)\sqrt{\frac{1+bN}{1+b}} + \lambda^{-1/2}\sqrt{2(1+bN)\big(\gamma_T^{\mathrm{st}} + \ln(N/\delta)\big)}\right)\sqrt{\lambda\,T\,\gamma_T^{\mathrm{mt}}(b)}
$$
$$
\leq 6\left(B\frac{1+b\epsilon}{1+b}\sqrt{\frac{(1+bN)(b+N)}{N}} + \sqrt{(1+bN)\big(\gamma_T^{\mathrm{st}} + \ln(N/\delta)\big)}\right)\sqrt{T\left(N\gamma_T^{\mathrm{st}} - \frac{Nb}{8(1+b)}\right)},
$$

where we have used the first claim of Proposition 1 and the choice $\lambda = (N+b)/(N+bN)$. Substituting $b = 0$ in the above equation, we recover the independent learning bound, i.e.,

$$
R^{\mathrm{mt}}(T) \leq 6\left(B\sqrt{NT\gamma_T^{\mathrm{st}}} + \sqrt{NT\gamma_T^{\mathrm{st}}\big(\gamma_T^{\mathrm{st}} + \ln(N/\delta)\big)}\right). \tag{42}
$$

Hence, even in the least favorable cases, the multitask approach allows to recover the independent baseline by using $b = 0$. Note that this is only made possible by the fact that $\beta_t^{\mathrm{small}\text{-}b}$ is tight at $b = 0$.

**A first bound for small $\epsilon$.** Here, we instead use the bound

$$
R^{\mathrm{mt}}(T) \leq 4\,\beta_T^{\mathrm{naive}}(b)\sqrt{\lambda\,T\,\gamma_T^{\mathrm{mt}}(b)}
$$
$$
= 4\left(B\sqrt{N(1+b\epsilon^2)} + \lambda^{-1/2}\sqrt{2\big(\gamma_T^{\mathrm{mt}}(b) + \ln(1/\delta)\big)}\right)\sqrt{\lambda\,T\,\gamma_T^{\mathrm{mt}}(b)}
$$
$$
\leq 4\left(B\sqrt{\frac{(1+b\epsilon^2)(b+N)}{1+b}} + \sqrt{2\left(\gamma_T^{\mathrm{st}} + \frac{NT}{b} + \ln(1/\delta)\right)}\right)\sqrt{T\left(\gamma_T^{\mathrm{st}} + \frac{NT}{b}\right)},
$$

where we have used the second claim of Proposition 1 and the choice $\lambda = (N + b)/(N + bN)$. Substituting $b = 1/\epsilon^2$ in the above equation, we obtain

$$
\begin{aligned}
\frac{R^{\mathrm{mt}}(T)}{6} &\leq \left( B\sqrt{\frac{N + 1/\epsilon^2}{1 + 1/\epsilon^2}} + \sqrt{\gamma_T^{\mathrm{st}} + \epsilon^2 NT + \ln(1/\delta)} \right) \sqrt{T\left(\gamma_T^{\mathrm{st}} + \epsilon^2 NT\right)} \\
&\leq \left( B\left(1 + \epsilon\sqrt{N}\right) + \sqrt{\gamma_T^{\mathrm{st}} + \epsilon^2 NT + \ln(1/\delta)} \right) \sqrt{T\left(\gamma_T^{\mathrm{st}} + \epsilon^2 NT\right)} \\
&\leq \sqrt{T\gamma_T^{\mathrm{st}}}\left( B + \sqrt{\gamma_T^{\mathrm{st}} + \ln(1/\delta)} \right) + \epsilon\left[ B\sqrt{NT\gamma_T^{\mathrm{st}}} + T\sqrt{N\gamma_T^{\mathrm{st}}} + B\sqrt{N}T \right. \\
&\qquad\qquad\qquad\qquad\qquad \left. + \epsilon BNT + \epsilon NT^{3/2} + T\sqrt{N\left(\gamma_T^{\mathrm{st}} + \ln(1/\delta)\right)} \right] \\
&= \sqrt{T\gamma_t^{\mathrm{st}}}\left( B + \sqrt{\gamma_T^{\mathrm{st}} + \ln(1/\delta)} \right) + \mathcal{O}\left( \epsilon\, BT\sqrt{N\left(\gamma_T^{\mathrm{st}} + \ln(1/\delta)\right)} + \epsilon^2 BNT^{3/2} \right).
\end{aligned}
$$
(43)

Bound (43) is composed of two terms: the single-task regret, and an additional term which scales with the deviation $\epsilon$ to the average task. Hence, as $\epsilon$ goes to 0, i.e., when tasks get more similar, we adaptively recover the single-task bound. Moreover, note that (43) is smaller than (42) as long as $\epsilon \leq 1/(N^{1/4}\sqrt{T})$. Hence, by choosing $b = (1/\epsilon^2) \cdot \mathbb{1}\{\epsilon \leq 1/(N^{1/4}\sqrt{T})\}$, we can obtain the minimum of the two bounds.

**A second bound for small $\epsilon$.**   Finally, we can use that

$$
\begin{aligned}
&R^{\mathrm{mt}}(T) \\
&\leq 4\,\beta_T^{\mathrm{large}\text{-}b}(b)\sqrt{\lambda\, T\, \gamma_T^{\mathrm{mt}}(b)} \\
&= 4\left( B\sqrt{\frac{(1 + b\epsilon)^2}{1 + b} + \frac{2bN}{1 + b} + \frac{2b(1 + b\epsilon)^2}{N\lambda^2(1 + b)^3}T^2} + \lambda^{-1/2}\sqrt{2\left(\gamma_T^{\mathrm{mt}}(b) + \ln(1/\delta)\right)} \right) \sqrt{\lambda\, T\, \gamma_T^{\mathrm{mt}}(b)} \\
&\leq 6\left( B\sqrt{\frac{(1 + b\epsilon)^2(b + N)}{(1 + b)^2 N} + \frac{2b(b + N)}{(1 + b)^2}\left(1 + \frac{(1 + b\epsilon)^2}{(b + N)^2}T^2\right)} + \sqrt{\gamma_T^{\mathrm{st}} + \frac{NT}{b} + \ln(1/\delta)} \right) \\
&\qquad \cdot \sqrt{T\left(\gamma_T^{\mathrm{st}} + \frac{NT}{b}\right)},
\end{aligned}
$$

where we have used the second claim of Proposition 1 and the choice $\lambda = (N + b)/(N + bN)$. Choosing $b = N/\epsilon^2$, we have

$$
\frac{(1 + b\epsilon)^2(b + N)}{(1 + b)^2 N} = \frac{\left(1 + \frac{N}{\epsilon}\right)^2\left(1 + \frac{1}{\epsilon^2}\right)}{\left(1 + \frac{N}{\epsilon^2}\right)^2} \leq \frac{2\left(1 + \frac{N^2}{\epsilon^2}\right)}{\left(1 + \frac{N}{\epsilon^2}\right)}\frac{1 + \epsilon^2}{N + \epsilon^2} \leq 2N\frac{5}{N} = 10\,,
$$

$$
\frac{2b(b + N)}{(1 + b)^2} \leq \frac{2(b + N)}{(1 + b)} = 2\frac{N + \frac{N}{\epsilon^2}}{1 + \frac{N}{\epsilon^2}} = 2N\frac{1 + \epsilon^2}{N + \epsilon^2} \leq 10\,,
$$

$$
\frac{1 + b\epsilon}{b + N} = \frac{b\epsilon + N\epsilon - N\epsilon + 1}{b + N} \leq \epsilon + \frac{1}{\frac{N}{\epsilon^2} + N} \leq \epsilon + \frac{\epsilon^2}{N} \leq 3\epsilon\,.
$$

Substituing in the above bound, we obtain

$$
\begin{aligned}
R^{\mathrm{mt}}(T) &= \mathcal{O}\left( B\sqrt{1 + \epsilon^2 T^2} + \sqrt{\gamma_T^{\mathrm{st}} + \epsilon^2 T + \ln(1/\delta)} \right) \sqrt{T\left(\gamma_T^{\mathrm{st}} + \epsilon^2 T\right)} \\
&= \mathcal{O}\left( \sqrt{T\gamma_T^{\mathrm{st}}}\left( B + \sqrt{\gamma_T^{\mathrm{st}} + \ln(1/\delta)} \right) + \epsilon BT^{3/2}\sqrt{\gamma_T^{\mathrm{st}}} \right. \\
&\qquad\qquad \left. + \epsilon T\left( B\sqrt{1 + \epsilon^2 T^2} + \sqrt{\gamma_T^{\mathrm{st}} + \epsilon^2 T + \ln(1/\delta)} \right) \right)
\end{aligned}
$$

---

**Algorithm 3** `AdaMT-UCB`

---

**Require:** Finite set $\mathcal{E} \subset (0, 2]$, learning rate $\eta > 0$.
 1: For $e \in \mathcal{E}$, initialize learner `MT-UCB`$(e)$ with $b, \lambda, \{\beta_t\}_{t \in \mathbb{N}}$ according to Theorem 2 using $\epsilon = e$.
 2: Initialize $\tau = U = R = 0$, $L^e = 0, \forall e \in \mathcal{E}$.
 3: **for** t=1,..., **do**
 4:  Choose learner $e_t = \min_{\epsilon \in \mathcal{E}}$.
 5:  Observe $i_t$ and play action from `MT-UCB`$(e_t)$, i.e. $x_t = \arg\max_{x \in \mathcal{X}} \text{ucb}_{t-1}^{e_t}(i_t, x)$.
 6:  Observe: $y_t = f^{\text{mt}}(i_t, x_t) + \xi_t$, and update *all* learners $\{\text{MT-UCB}(e)\}_{e \in \mathcal{E}}$ based on observation.
 7:  Accumulate:

$$\tau \mathrel{+}= 1, \quad U \mathrel{+}= y_t, \quad R \mathrel{+}= 2\beta_{t-1}^{e_t}\sigma_{t-1}^{e_t}(i_t, x_t), \quad L^e \mathrel{+}= \text{lcb}_{t-1}^e(i_t, x_t), \forall e \in \mathcal{E}.$$

 8:  Misspecification test: $\qquad U + R + c\sqrt{\tau \ln(\ln(\tau)/\delta)} < \max_{e \in \mathcal{E}} L^e \qquad\qquad$ (46)

 9:  **if** condition (46) is true **then**
10:    # Learner $e_t$ is misspecified w.h.p.. Hence, start a new epoch.
11:    $\mathcal{E} = \mathcal{E} \setminus \{e_t\}$ and reset $\tau = U = R = 0$, $L^e = 0, \forall e \in \mathcal{E}$.

---

$$= \mathcal{O}\left(\sqrt{T\gamma_T^{\text{st}}}\left(B + \sqrt{\gamma_T^{\text{st}} + \ln(1/\delta)}\right) + \epsilon BT^{3/2}\sqrt{\gamma_T^{\text{st}} + \ln(1/\delta)} + \epsilon^2 BT^2\right). \quad (44)$$

Again, bound (44) is composed of two terms: the single-task regret, and an additional term which goes to 0 as $\epsilon$ goes to 0. Interestingly, this last part is independent from $N$, which is a consequence of $\beta_t^{\text{large-}b}$ being $\sqrt{N}$ smaller than $\beta_t^{\text{naive}}$ at $b = +\infty$. For small values of $T$, namely when $T \leq N$, (44) is thus smaller than (43). Choosing $b = N/\epsilon^2$ when $T \leq N$, and as before otherwise, ensures to obtain the minimum of (42), (43), and (44). $\qquad\qquad\square$

## B.4 Adapting to unknown tasks' similarity

In Algorithm 3 we summarize the `AdaMT-UCB` approach discussed in Section 4. In the misspecification test (Line 8), $\text{lcb}_t^e$ are lower confidence bound functions, defined as:

$$\text{lcb}_t^e(i, x) = \mu_t(i, x \mid b^e) - \beta_t(b^e) \cdot \sigma_t(i, x \mid b^e),$$

where $b^e$ is the kernel parameter chosen (according to Theorem 2) by each learner $e$. Moreover, $c$ is an absolute constant such that, by standard concentration arguments and for all times $\tau$,

$$\left| U - \sum_{t=1}^{\tau} f^{\text{mt}}(i_t, x_t) \right| \leq c\sqrt{\tau \ln(\ln(\tau)/\delta)}, \qquad\qquad (45)$$

with probability $1 - \delta$, see, e.g., [22, Lemma B.1].

### B.4.1 Proof of Theorem 3

**Theorem 3.** *Assume that there exists $e \in \mathcal{E}$ such that $e \geq \epsilon$, and let $M$ be the number of learners $e \in \mathcal{E}$ such that $e < \epsilon$ (i.e., the number of misspecified learners in $\mathcal{E}$). The regret of* `AdaMT-UCB` *satisfies with high probability $R^{\text{mt}}(T) = \mathcal{O}\big(\sqrt{M+1} \cdot \overline{R_\star^{\text{mt}}}(T)\big)$.*

Among the set of learners defined by parameters $e \in \mathcal{E}$, we have identified with $e^\star$ the (well-specified) learner with the smallest $e$ such that $e \geq \varepsilon$. Then, our goal is to obtain a regret bound which grows as the regret of learner $e^\star$. We have denoted with $\overline{R_\star^{\text{mt}}}(T)$ the regret bound of such learner had it been chosen from time 0.

We first prove the following auxiliary lemma, which is the analog of [22, Theorem 7.1].

**Lemma 3.** *With probability at least $1 - \delta$, the misspecification test in Equation (46) does not trigger if all learners in $\mathcal{E}$ are well-specified and their confidence intervals contain $f^{\text{mt}}$.*

*Proof.* When all learners $e \in \mathcal{E}$ are well-specified and their intervals contain the true function, for all $t$ it holds $\text{lcb}_t^e(i_t, x_t) \leq \max_{x \in \mathcal{X}} f^{\text{mt}}(i_t, x)$. Thus, for each learner $e \in \mathcal{E}$, with probability at least

$1 - \delta$,

$$L^e = \sum_{t=1}^{T} \mathrm{lcb}_{t-1}^e(i_t, x_t)$$

$$\leq \sum_{t=1}^{T} \max_{x \in \mathcal{X}} f^{\mathrm{mt}}(i_t, x)$$

$$\leq \sum_{t=1}^{T} f^{\mathrm{mt}}(i_t, x_t) + 2\beta_{t-1}^{e_t}\sigma_{t-1}^{e_t}(i_t, x_t)$$

$$\leq U + c\sqrt{T \ln(\ln(T)/\delta)} + R,$$

where, in addition to Equation (45), we have used that $\sum_t \max_{x \in \mathcal{X}} f^{\mathrm{mt}}(i_t, x) - f^{\mathrm{mt}}(i_t, x_t) \leq \sum_t \mathrm{ucb}_t(i_t, x_t) - \mathrm{lcb}_t(i_t, x_t) = \sum_t 2\beta_{t-1}^{e_t}\sigma_{t-1}^{e_t}(i_t, x_t)$. Thus, the misspecification test of (46) does not trigger. $\qquad\square$

Let us now bound the overall regret of `AdaMT-UCB`. First, we can decompose it into the regrets inside each epoch:

$$R^{\mathrm{mt}}(T) = \sum_{t=1}^{T} \max_{x \in \mathcal{X}} f^{\mathrm{mt}}(i_t, x) - f^{\mathrm{mt}}(i_t, x_t)$$

$$= \sum_{s=1}^{\text{\# of Epochs}} \sum_{t \in \text{Epoch-}s} \max_{x \in \mathcal{X}} f^{\mathrm{mt}}(i_t, x) - f^{\mathrm{mt}}(i_t, x_t)$$

$$= \sum_{s=1}^{\text{\# of Epochs}} R_s^{\mathrm{mt}}(T_s),$$

where we have defined $T_s$ to be the duration of epoch $s$ and $R_s^{\mathrm{mt}}(T_s)$ its corresponding regret. Note that by Lemma 3, the maximum number of terminated epochs corresponds to the number of misspecified learners in the initial set $\mathcal{E}$. Thus, letting $M$ be such number, with high probability:

$$R^{\mathrm{mt}}(T) \leq \sum_{s=1}^{M+1} R_s^{\mathrm{mt}}(T_s). \tag{47}$$

**During each epoch.** Let us now look at what happens during each epoch $s$. For simplicity, we will condition on the event that the intervals of learner $e^\star$ contain the true $f^{\mathrm{mt}}$, and on the event of Equation (45). Note that by definition, during each epoch the misspecification test has not triggered. In particular, this is true when testing against learner $e^\star$. That is,

$$U + R + c\sqrt{T_s \ln(\ln(T_s)/\delta)} \geq \sum_{t \in \text{Epoch-}s} \mathrm{lcb}_{t-1}^{e^\star}(i_t, x_t),$$

which, by letting $e_s = \min_{e \in \mathcal{E}}$ be the learner utilized in epoch $s$, implies:

$$\sum_{t \in \text{Epoch-}s} f^{\mathrm{mt}}(i_t, x_t) + 2\beta_{t-1}^{e_s}\sigma_{t-1}^{e_s}(i_t, x_t) + 2c\sqrt{T_s \ln(\ln(T_s)/\delta)} \geq \sum_{t \in \text{Epoch-}s} \mathrm{lcb}_{t-1}^{e^\star}(i_t, x_t). \tag{48}$$

Then, using the above condition:

$$R_s^{\mathrm{mt}}(T_s) - 2c\sqrt{T_s \ln \frac{\ln(T_s)}{\delta}}$$

$$= \sum_{t \in \text{Epoch-}s} \max_{x \in \mathcal{X}} f^{\mathrm{mt}}(i_t, x) - f^{\mathrm{mt}}(i_t, x_t) - 2c\sqrt{T_s \ln \frac{\ln(T_s)}{\delta}}$$

$$\text{(By Eq. (48))} \quad \leq \sum_{t \in \text{Epoch-}s} \max_{x \in \mathcal{X}} f^{\mathrm{mt}}(i_t, x) - \mathrm{lcb}_{t-1}^{e^\star}(i_t, x_t) + \sum_{t \in \text{Epoch-}s} 2\beta_{t-1}^{e_s}\sigma_{t-1}^{e_s}(i_t, x_t)$$

$(e^\star \text{ is well-specified}) \qquad \leq \sum_{t \in \text{Epoch-}s} 2\beta_{t-1}^{e^\star} \sigma_{t-1}^{e^\star}(i_t, x_t) + \sum_{t \in \text{Epoch-}s} 2\beta_{t-1}^{e_s} \sigma_{t-1}^{e_s}(i_t, x_t)$

$$\leq 4\,\beta_{T_s}(b^{e^\star})\sqrt{\lambda(b^{e^\star})\,T_s\,\gamma_{T_s}^{\text{mt}}(b^{e^\star})} + 4\,\beta_{T_s}(b^{e_s})\sqrt{\lambda(b^{e_s})\,T_s\,\gamma_{T_s}^{\text{mt}}(b^{e_s})}$$

$$\leq 2\overline{R_\star^{\text{mt}}}(T_s)\,.$$

In the last two inequalities, we have used the same proof steps of Lemma 1 to bound the sum of confidence widths for learners $e^\star$ and $e_s$, and finally, the fact that $e_s \leq e^\star$ and thus the regret bound of learner $e_s$ is bounded by $\overline{R_\star^{\text{mt}}}(T_s)$ (since the bound from Theorem 2 increases with $\epsilon$).

Overall, combining the latter with Equation (47), we obtain

$$R^{\text{mt}}(T) \leq \sum_{s=1}^{M+1} 2\overline{R_\star^{\text{mt}}}(T_s) + 2c\sqrt{T_s \ln \frac{\ln(T_s)}{\delta}}$$

$$= 2\sum_{s=1}^{M+1} C^\star(T_s)\sqrt{T_s} + c\sqrt{T_s \ln \frac{\ln(T_s)}{\delta}}$$

$$\leq 2C^\star(T)\sqrt{T}\sqrt{M+1} + 2c\sqrt{(M+1)T \ln \frac{\ln(T)}{\delta}}$$

$$= 2\overline{R_\star^{\text{mt}}}(T)\sqrt{M+1} + 2c\sqrt{(M+1)T \ln \frac{\ln(T)}{\delta}}\,.$$

where we have use the fact that MT-UCB regret bounds are of the form $\overline{R_\star^{\text{mt}}}(T_s) = C^\star(T_s)\sqrt{T_s}$ for some appropriate function $C^\star(T_s)$, see Theorem 2, and Jensen's inequality. $\qquad\square$

**How many learners are needed?** Let $\mathcal{E}$ be the exponential grid $\{1, \rho, \rho^2, \ldots, \rho^{M-1}\}$, for some $\rho < 1$. Let $\epsilon \in [0,2]$ be the true tasks similarity parameter. By definition, the best learner is better than the learner $m^*$ such that $\rho^{m^*+1} \leq \epsilon \leq \rho^{m^*}$. The estimate it uses for $\epsilon$ is better than $\epsilon^* := \rho^{m^*}$, which satisfies $\epsilon/\epsilon^* \in [\rho, 1]$. Hence, the bigger $\rho$, the more precise we are. However, the number of learners needed also increases with $\rho$. Indeed, if we want to be able to identify up to $\epsilon_{\min}$, we have to choose $M$ such that

$$\rho^{M-1} \leq \epsilon_{\min} \qquad \text{or again} \qquad M \geq 1 + \frac{\log(1/\epsilon_{\min})}{\log(1/\rho)}\,.$$

## C  Active Learning

### C.1  Proof of Theorem 4

**Theorem 4.** *Suppose that for all tasks $i$, point $x$, and time $t$, we have that $f^{\text{mt}}(i,x) \in [\mu_t(i,x) \pm \beta_t^i \cdot \sigma_t(i,x)]$. Then, the* MT-AL *algorithm ensures the active learning regret is bounded by*

$$R_{\text{AL}}^{\text{mt}} \leq 2\sum_{t=1}^{T} \beta_t^{i_t} \sigma_t(i_t, x_t^{i_t})\,,$$

*where $\{i_t\}$ is the sequence of queried tasks and $\{x_t^{i_t}\}$ the strategies selected for each of them.*

*Proof.* Let $x_\star^i \in \arg\max_{x \in \mathcal{X}} f^{\text{mt}}(i,x)$. Then, the active learning regret of MT-AL can be bounded as

$$R_{\text{AL}}^{\text{mt}}(T) := \sum_{t=1}^{T} \frac{1}{N} \sum_{i=1}^{N} f^{\text{mt}}(i, x_\star^i) - \sum_{t=1}^{T} \frac{1}{N} \sum_{i=1}^{N} f^{\text{mt}}(i, x_t^i)$$

$$\leq \sum_{t=1}^{T} \frac{1}{N} \sum_{i=1}^{N} \text{ucb}_{t-1}(i, x_\star^i) - \sum_{t=1}^{T} \frac{1}{N} \sum_{i=1}^{N} \text{ucb}_{t-1}(i, x_t^i) + 2\sum_{t=1}^{T} \frac{1}{N} \sum_{i=1}^{N} \beta_{t-1}^i \sigma_{t-1}(i, x_t^i)$$

$$\leq 2\sum_{t=1}^{T} \frac{1}{N} \sum_{i=1}^{N} \beta_{t-1}^i \sigma_{t-1}(i, x_t^i)$$

$$\leq 2 \sum_{t=1}^{T} \beta_{t-1}^{i} \sigma_{t-1}(i_t, x_t^{i_t}) \underbrace{\frac{1}{N} \sum_{i=1}^{N} 1}_{=1}$$

The first inequality holds since by assumption, for all tasks $i$, point $x$, and time $t$, we have $f^{\mathrm{mt}}(i, x) \in [\mu_t(i, x) \pm \beta_t^i \cdot \sigma_t(i, x)]$. The second one follows since, at each round $t$ MT-AL select $x_t^i = \arg\max_x \mathrm{ucb}_{t-1}(i, x)$ for all $i$, and the third one since $i_t = \arg\max_i \beta_{t-1}^i \sigma_{t-1}(i, x_t^i)$. $\qquad\square$

### C.2 Proof of Corollary 2

**Corollary 2.** *Let* MT-AL *utilize the MT regression estimates of Eq. (3)-(4) with parameters set according to Theorem 2. Moreover, let $\overline{R^{\mathrm{mt}}}(T)$ be the bound on the online learning regret obtained in Theorem 2. Then, with high probability, we have $R_{\mathrm{AL}}^{\mathrm{mt}}(T) \leq \overline{R^{\mathrm{mt}}}(T)$.*

*Proof.* First, since MT-AL utilizes the MT regression estimates of Eq. (3)-(4) with parameters set according to Theorem 2, with high probability $f^{\mathrm{mt}}(i, x) \in [\mu_t(i, x) \pm \beta_t^i \cdot \sigma_t(i, x)]$ and the results of Theorem 4 holds. Then, the result follows since, according to the proof of Theorem 2, for every sequence of revealed tasks $\{i_t\}_{t=1}^{T}$ (in particular the ones chosen by MT-AL), it holds $2\sum_{t=1}^{T} \beta_{t-1}^i \sigma_{t-1}(i_t, x_t^{i_t}) \leq \overline{R^{\mathrm{mt}}}(T)$, see Appendix B. $\qquad\square$

### C.3 Comparison with AE-LSVI [18]

The proposed MT-AL algorithm can be compared with the offline contextual Bayesian algorithm AE-LSVI [18] whose goal is to quickly discover the optimal strategy for each context (task, in our case). Like MT-AL, AE-LSVI selects strategy $x_t^i = \arg\max_x \mathrm{ucb}_{t-1}(i, x)$ for each task $i$. However, unlike MT-AL, AE-LSVI queries the task $i_t = \arg\max_{i \in [N]}[\mathrm{ucb}_t(i, x_t^i) - \max_{x \in \mathcal{X}} \mathrm{lcb}_t(i, x)]$. This, together with a terminal rule for the final reported actions $\{x_T^i\}_{i=1}^{N}$, ensures the latter are $\mathcal{O}(\beta_T \sqrt{\gamma_T^{\mathrm{mt}}}/\sqrt{T})$-approximate optimal for each task. We note that a similar error can also be proven when using MT-AL, by turning the active learning regret bound into a last-iterate approximation error (effectively dividing the regret by $T$). However, unlike our approach, it is unclear whether the active learning regret of AE-LSVI is sublinear. In particular, by querying the task with maximal uncertainty MT-AL controls the regrets for all the other tasks (see last inequality in Proof of Theorem 4). Instead, the query strategy of AE-LSVI considers a truncated uncertainty, since $[\mathrm{ucb}_t(i, x_t^i) - \max_{x \in \mathcal{X}} \mathrm{lcb}_t(i, x)] \leq [\mathrm{ucb}_t(i, x_t^i) - \mathrm{lcb}_t(i, x_t^i)] = \beta_t^i \sigma_t(i, x_t^i)$.

## D Additional experimental results

All our experiments were run using 8 CPUs at 3.7 GHz.

In Figure 4 we report online learning an active learning regrets of additional synthetic experiments for different values of $N$ (number of tasks) and parameter $\delta$ (inversely proportional to the tasks' similarity, see Section 5). We observe that the improvement of MT regression over independent learning increases with the number of tasks and with their similarity (i.e., for small $\delta$). Moreover, as $N$ increases, the improved confidence intervals further improve over the naive ones, as well as the benefit of active learning compared to uniform sampling. All these considerations conform with our theory.

In our active learning experiments, we further analyze the frequencies of each task being queried by the considered baselines. In Figure 5, we report such frequencies averaged over 10 runs for a random instance of our synthetic experiments with $N = 5$ and $\delta = 0.4$, i.e. the setup of Figure 3 (b). We observe the active learning baselines (MT-AL, MTS, and AE-LSVI) query more frequently task-1 and task-5, which are found to be the tasks with the smallest norm $\|f_i\|$ and thus with the smallest signal-to-noise ratio. In Figure 6, we report the query frequencies for our drug discovery experiments averaged over 5 runs. We observe MT-AL and AE-LSVI (which both query tasks with maximal uncertainty) query slightly more often task-3 (corresponding to allele A-0203). Instead, MTS (which query the task yielding the maximum performance improvement) focuses significantly more often task-1 (allele A-0201). This produces high rewards in the early rounds but incurs linearly growing regret overall (see Figure 3 (b)).

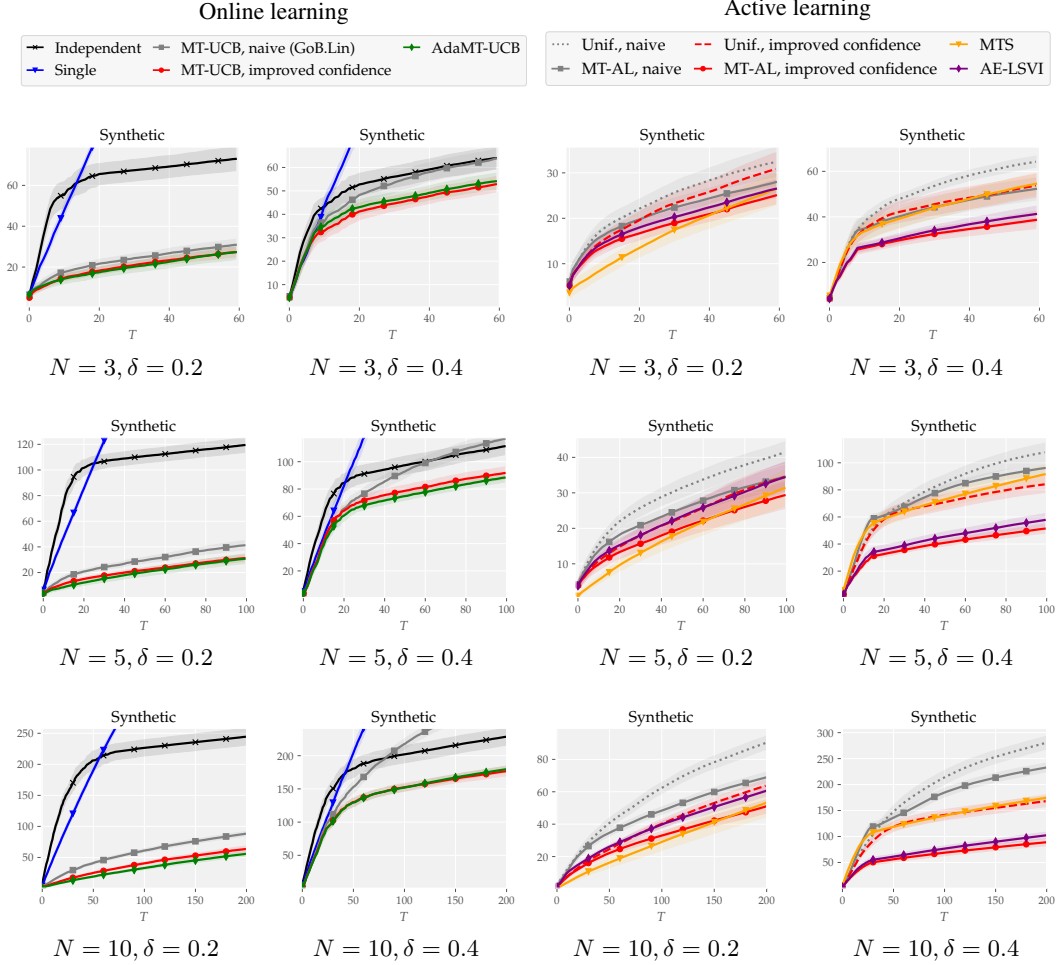

Figure 4: Online (left) and active (right) learning regrets of synthetic experiments for different parameters: $N$ is the number of tasks while parameter $\delta$ is inversely proportional to the tasks' similarity, see Section 5 for how task vectors are generated.

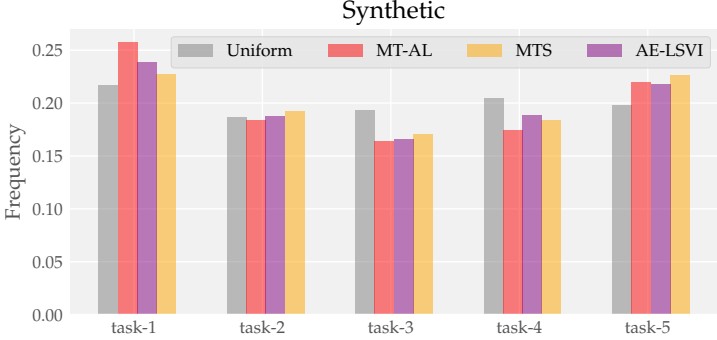

Figure 5: Frequency each task being queried by each of the considered baselines. The plot refers to our synthetic experiments with $N = 5, \delta = 0.4$, i.e. the setup of Figure 3 (b). We generate a single set of task vectors $\{f_i\}_{i=1}^N$ and average results over 10 runs, each with horizon $T = 100$.

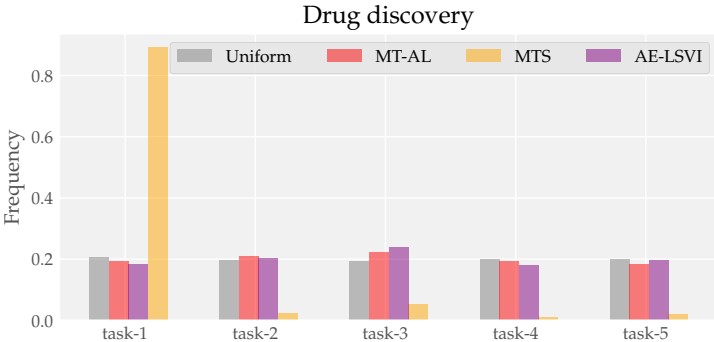

Figure 6: Frequency of each task being queried by each of the considered baselines for our drug discovery experiments (results averaged over 5 runs, each with horizon $T = 600$).