# OpenReview forum: "Multitask Learning with No Regret: from Improved Confidence Bounds to Active Learning"
_NeurIPS.cc/2023/Conference — NeurIPS 2023 poster_

### Official Review · Reviewer_w9gE · 2023-06-21

**Soundness:** 3 good
**Presentation:** 3 good
**Contribution:** 3 good
**Rating:** 7
**Confidence:** 5

**Summary:**

This paper proposes refined confidence intervals for multitask kernel regression and applies this to the multitask online learning setting and the multitask active learning setting. The thrust of the idea is to further exploit the structure of the task kernel (Eq (2)) where the refined confidence bound is always smaller than the existing confidence bound and tighter by a factor of sqrt of the number of tasks for the extreme cases of b=0 and b=infty, corresponding to the independent tasks or identical tasks, respectively.

Two important parameters: $b$ is the task-relatedness in the definition of the multitask kernel and $\eps$ is the task discrepancy from the average task.

The paper studies both cases of known and unknown $\eps$. When $\eps$ is unknown, they use a minibatch learning idea and misspecification task to determine the smallest upper bound of $\eps$ from the predefined search space. This only incurs an extra term of the square root of the search space cardinality.

The paper also provides empirical validations to justify their results.

----------
UPDATE: I am raising my rating from 5 to 7 to champion this paper. I think this is a solid contribution to the multitask learning setting and I would view this paper as an (interesting) theoretical contribution, rather than a strong algorithm paper in the deep learning setting. On this ground, I think the experimental results the paper presented are sufficient for validating the usefulness of multitask learning in the settings they considered. I believe the points raised by Reviewer ckqH are important but can be largely addressable without changing the scientific merit claimed in the reviewed paper. In particular, I agree with Reviewer ckqH in that the paper should make it clear from the title and the abstract that the paper focuses on a multitask kernel setting, not a general multitask learning setting.

**Strengths:**

- a new refined confidence bound that is tighter than the naive bound by exploiting the structure of multitask kernel. This result is neat.
- a comprehensive study where they apply to various learning settings and provide empirical results

**Weaknesses:**

- some of the writing is not clear but improbable
- the notion of regret in section 4 looks weird

**Questions:**

- An implicit assumption is that each $f_t$ is a sample of the known $GP(k_X)$ that I feel the paper does not make it explicit enough.
- Also the task kernel in Eq (2) driven by $b$ is a modeling choice, not an assumption because the only task relatedness assumption comes from $\epsilon$. The author can correct me on this point if I'm wrong. Either way, whether the task kernel is an assumption or a modeling choice should be made explicitly to the readers.
- Line 211: $N >= T$ instead?
- Line 248 - 250 about the misspecification test: the intuition behind why this test works should be explained further (I know what the test means but its current explanation is not very helpful to all readers).
- The regret notion in Section 4 looks weird to me. Isn't $i_t$ instead of $i$ in the second term of the regret notion? Because the learner actively queries a task, thus, the goodness of the queried tasks should be included in the regret.
- The second and third bound in Theorem 2 is definitely helpful when $\epsilon = 0$. In which other regimes they are also helpful? and which are not?

**Limitations:**

Yes.

---

> ### Author Rebuttal · Authors · 2023-08-07
>
> We thank the reviewer for raising these questions, which we address below.
>
> > An implicit assumption is that each $f_t$  is a sample of the known $GP(k_X)$  that I feel the paper does not make it explicit enough.
>
> In this work, we take a frequentist interpretation and assume the tasks$ f_i$ are fixed (but unknown) functions in the reproducing kernel Hilbert space of a known kernel $k_x$. We mention such an assumption at the beginning of Section 2.1. This is the frequentist analog of the Gaussian process assumption mentioned by the reviewer (see, e.g., [Srinivas et al. 2010] where both Bayesian and frequentist interpretations are compared), and allows one to learn such functions via kernel regression.
>
>
> > Also the task kernel in Eq (2) driven by $b$ is a modeling choice, not an assumption because the only task relatedness assumption comes from $\epsilon$. The author can correct me on this point if I'm wrong. Either way, whether the task kernel is an assumption or a modeling choice should be made explicitly to the readers.
>
> The reviewer is perfectly correct about the task kernel in Eq. (2) being a modeling choice (and not an assumption) and indeed it is what allows one to exploit the benefits of multitask learning. The choice of $b$ can interpolate between single-task and all-tasks regression (as we show in Proposition 2 in Appendix A.1). Indeed, the derived intervals (see, e.g., Figure 2) are a function of the parameter $b$ that one decides to use. We will make sure it comes across clearly. In Theorem 2 –in particular– we suggest a possible theoretical choice for $b$ (as a function of $N$, $T$, and $\epsilon$) which leads to the derived online learning regret bounds.
>
>
> > Line 211: $N \ge T$ instead?
>
> Note that the second bound in Theorem 2 grows with $T^{3/2}$, while the third bound with $T\sqrt{N}$. Thus, the latter is beneficial (i.e., smaller) when $N \leq T$.
>
>
> > Line 248 - 250 about the misspecification test: the intuition behind why this test works should be explained further.
>
> We acknowledge that the intuition and the role of the misspecification test can be explained further. Upon acceptance, we would use the extra page to move the AdaMT-UCB algorithm (currently in Appendix) to the main text and provide extra intuition about the test.
>
>
> > The regret notion in Section 4 looks weird to me. Isn't $i_t$ instead of $i$ in the second term of the regret notion? Because the learner actively queries a task, thus, the goodness of the queried tasks should be included in the regret.
>
> The fact that the active learning regret of Section 4 depends on all tasks $i$ and *not* exclusively on the queried one $i_t$ is precisely what makes it different and more challenging than the standard online learning regret of Eq. (6). Indeed if in online learning we only care about the task which is revealed at each round (i.e., $i_t$), in active learning we care *simultaneously* about all the tasks (hence, the regret depends on all $i \in [N]$ at each round) and face the additional challenge of deciding which task to query to learn them as fast as possible. If the regret were only dependent on the queried task $i_t$, then a trivial good strategy would be to always query the ``easiest” one, yielding a small regret but not learning the other $N-1$ tasks. We hope this clarifies the reviewer’s perplexity.
>
> > The second and third bound in Theorem 2 is definitely helpful when $\epsilon = 0$. In which other regimes they are also helpful? and which are not?
>
> Compared to the first bound (which corresponds to independent learning), the second and third bounds in Theorem 2 refer to multitask learning. As hinted by the reviewer, they are always beneficial for $\epsilon=0$ (i.e. when tasks are all identical) but when $\epsilon>0$ their benefit generally depends on the interplay among the number of tasks $N$, the horizon $T$, and $\epsilon$ itself. As discussed after Theorem 2, for a given $\epsilon$ the benefit of such bounds degrades as $T$ increases (large data regimes) or when the number of tasks $N$ is small. Moreover, note that the second bound does not depend *at all* on $N$. Thus, for any $\epsilon$ and $T$ there exists a number of tasks $N$ above which multitask learning is beneficial. Finally, we remark that the particular choice of $b$ derived in Theorem 2 allows MT-UCB to always attain *the minimum* of such three bounds and thus it can never be worse than independent learning.
> \
> \
> \
> We hope the above points address the reviewer’s concerns and we are happy to provide additional insights if needed.

---

> > ### Comment · Reviewer_w9gE · 2023-08-14
> >
> > I thank the authors for answering my questions. It clarified most of my questions, except for the regret notion in Section 4. The learner deciding which tasks to query is I still think precisely why we need to measure the goodness of the queried tasks $i_t$ against all the tasks $i \in [N]$. The existing regret notion does not capture the goal of learning **each** tasks well. For example, the existing regret can still be small even when the learner learns $T-1$ "dominating" tasks very well but learns nothing about the other task. Anyhow, this section is an application of the main results in the paper and I tend to keep my original rating.

---

> > > ### Author Response · Authors · 2023-08-14
> > >
> > > Thank you for the follow-up question.
> > > We agree with the reviewer that the notion of active learning regret $R_{AL}^\text{mt}(T)$ looks at the performance of the tasks “on average” and thus it may be small when the first $N-1$ tasks are much more optimized than task $N$. However, we note that in order for $R_\text{AL}^\text{mt}(T)$ to grow sublinearly, the learner must eventually discover the optimal strategy for each task. Otherwise, if task $N$ is “left behind”, its regret grows linearly with $T$ and so does $R_\text{AL}^\text{mt}(T)$. In general, our theory is also extendable to non-uniform importance over the tasks and to the “robust” objective hinted at by the reviewer, where the learner should be penalized for the worst task.
> > >
> > > We hope this clarifies the reviewer's concern and we are happy to explain further if needed.

---

### Official Review · Reviewer_ckqH · 2023-06-27

**Soundness:** 2 fair
**Presentation:** 1 poor
**Contribution:** 2 fair
**Rating:** 3
**Confidence:** 3

**Summary:**

This paper focuses on multi-task regression problems where they provide tighter confidence bounds that can be used to establish regret bounds in multi-task online learning and multi-task active learning. They also propose an algorithm that reduces the regret compared to previous bounds by adapting a task similarity parameter automatically. They perform regression experiments on a synthetic dataset and a drug discovery dataset (MHC-I) to show that their algorithms yield better (lower) regret than the reported baselines.

**Strengths:**

The proposed confidence and regret bounds are general in the sense that they can be applied in online and active learning in the multi-task setting. At times, the paper prepares the reader what a section with a summary sentence, e.g. on L86-87 “In this section, we introduce the MT kernel regression setting, and prove our refined confidence intervals.” The multi-task active learning setting where only one of the tasks can be queried per time step is interesting.

**Weaknesses:**

**W1:** I am not a multi-task learning expert, however, I have to say that I find the paper hard to follow. There is a lot of inline math which makes reading less smooth. Also, most of the notation is introduced within the sections, while a separate paragraph that thoroughly describes all the notation that will come would have been of great help for the reader to go back to if necessary. Furthermore, I think the paper needs re-writing to emphasize what the main messages are to the reader in each section and paragraph.

**W2:** One of the main contributions, Algorithm 3, described in Section 3.2 is placed in the Appendix, probably due to lack of space. I suggest that the Section 2 and 3 are re-written such that Algorithm 3 can fit into the main paper.

**W3:** Experiment section is very short (<1.5 pages). I believe that the paper would benefit from adding more experimental results to justify the contributions, as well as adding discussions around the results and emphasizing on the take-home messages to bring insights to the reader.

**W4:** In my opinion, the scope of the paper is narrow as it only covers regression tasks in multi-task learning without touching any classification tasks. Furthermore, there are no experiments using deep neural networks which makes me doubt how relevant this paper is for the general audience at Neurips.


**Questions:**

**Q1:** Can you elaborate on the interpretation of the expression for calculating “regret” in Eq. 6? And can you elaborate on what a “no-regret algorithm” is?

**Q2:** Can the proposed confidence intervals and regret bounds be used for classification tasks? Can they be applied for deep neural networks in multi-task learning problems for both regression and classification?

**Q3:** Are there any metrics other than regret  that are relevant to report in the experiments, e.g., RMSE?

**Q4:** Since you are focusing in this agnostic setting where the task similarity is unknown, I am wondering if you can mention any examples or applications where the task similarity parameter $b$ is actually known? I get the feeling that $b$ is unknown in most multi-task learning problems.


**Limitations:**

No limitations are addressed as this is a theoretical work according to the authors. See Weaknesses and Questions above for limitations that I am concerned over.

---

> ### Author Rebuttal · Authors · 2023-08-07
>
> We thank the reviewer for going through our work. We respond below to the raised weaknesses and questions.
>
> **W1.**
> We will use the extra page upon acceptance to introduce our notation in a separate paragraph.
>
>
>
> **W2.**
> We will use the extra page upon acceptance to move Algorithm 3 into the main text.
>
>
> **W3.**
> We would like to point out that our work is primarily of theoretical nature and that experiments are essentially designed to support our theoretical findings. In that respect, we believe our experimental setup provides empirical evidence to all of our following claims:
> - the novel confidence intervals derived in Theorem 1 outperform their naive counterparts (as depicted in Figures 1 and 2) and are key for effective multitask learning. This is confirmed both in our online and active learning experiments where we run baselines using the naive intervals and our improved ones. In addition, in our set of synthetic experiments (see Figure 4 in Appendix D) we observe that the higher the number of tasks $N$, the more our improved intervals outperform the naive ones. This perfectly conforms with –and experimentally verifies– our theory.
> - the algorithms we introduce (MT-UCB-improved confidence, AdaMT-UCB, MT-AL-improved confidence) outperform existing methods, including the independent single-task approach, see Figures 3 and 4.
> - AdaMT-UCB, which automatically adapts to the tasks relatedness, is at least on par (if not better) with the strategy that can access (a bound on) the tasks’ deviation $\epsilon$, see Figures 3 (a) and 4.
>
> In addition, following the suggestion of Reviewer J9Sp, we have performed additional analyses and interpretations for our active learning experiments reporting the frequencies of each task being queried by each of the baselines (see Figures in the pdf attached to the general rebuttal).
>
>
> **W4.**
> **We respectfully disagree.** First, regression is one of the most fundamental problems in machine learning; many ML algorithms build upon standard single-task regression and the associated confidence intervals. More recently –and among many other applications– these are used, e.g., in reinforcement learning, or in combination with deep models which provide powerful embeddings, thus enabling a whole set of downstream applications. Regarding the relevance to the venue, we would like to point out that deep learning is just one of the several topic areas at NeurIPS. Moreover, as discussed above, our experimental setup is specifically designed to support our theoretical findings and we do not need to use neural networks to do so.
>
>
> **Q1.**
> The notion of regret typically measures the difference between the performance achieved by the learner and that of the best comparator for the considered problem. This is the de-facto metric in online learning (and not only, see e.g. [1,2,3]), where it is used to quantify the performance of sequential decision-making algorithms. The learner is said to achieve “no-regret” if its regret $R(T)$ grows sublinearly with $T$ (or equivalently $R(T)/T \rightarrow 0$). Thus, as $T\rightarrow \infty$ its average performance converges to the one of the best comparator. In our multitask (MT) setting, in particular, the regret of Eq. (6) quantifies the cumulative difference between the best reward achievable for each revealed task and the reward of the strategies chosen by the learner. Thus, a no-regret MT algorithm is such that as $T\rightarrow \infty$ the learner discovers the best strategy for each task.
>
> [1]: Cesa-Bianchi, N., & Lugosi, G. (2006). Prediction, learning, and games. Cambridge university press.
>
> [2]: Agarwal, N., E. Hazan, and K. Singh (2019). "Logarithmic regret for online control." Advances in Neural Information Processing Systems 32.
>
> [3]: Auer, P., Jaksch, T., & Ortner, R. (2008). Near-optimal regret bounds for reinforcement learning. Advances in Neural Information Processing Systems, 21.
>
>
> **Q2.**
> They only apply to regression because they exploit the fact that the prediction space is continuous. However, they can be combined with deep networks, see e.g. [4], to perform linear regression on the last layer (i.e., the deep-net provides embeddings). This is becoming popular for uncertainty quantification and online learning.
>
> ​​[4] Damianou, A., & Lawrence, N. D. (2013, April). Deep Gaussian processes. In Artificial intelligence and statistics (pp. 207-215). PMLR.
>
>
> **Q3.**
> Because our work focuses on *sequential* decision-making setups, the regret is the main relevant performance indicator to quantify the speed of learning. RMSE better applies to offline regression in the presence of train/validation splits, which is not our focus.
>
>
>
> **Q4.**
> We would like to remind the reviewer that parameter $b$ does not correspond to the tasks’ similarity, but to a tunable modeling parameter that dictates how much information one wants to share across tasks (see, e.g. Proposition 2 in the Appendix). Indeed, our confidence intervals (and guarantees) hold true for the whole set of possible choices of $b$. However, though, some choices of $b$ perform better than others for the given problem. In Theorem 2 we have derived a theoretical choice for $b$ which optimizes the regret bound of MT-UCB up to constants. In practice, although we have tried different values and picked the best, one could run more sophisticated hierarchical approaches (e.g. based on experts) to learn the best $b$ online. We feel that this is a complementary approach beyond the scope of the paper. Note that the situation is different w.r.t. parameter $\epsilon$, which instead *must be* a known bound on the tasks’ deviation and for which we have designed the adaptive AdaMT-UCB procedure.

---

> > ### Comment · Reviewer_ckqH · 2023-08-11
> > **Reply to Authors: Concerns on presentation and potential impact**
> >
> > Thank you for responding to my concerns and questions.
> >
> > My main concerns are on the presentation and the potential impact of the paper. I congratulate the authors on their thorough job on the theoretical parts. However, the paper structure would benefit from making the transitions between sections and paragraphs more smooth to help the reader, where e.g. applying “topic sentences” is a good approach. Furthermore, the motivation why the proposed confidence bounds are useful could be made more clear by e.g. incorporating real-world examples. More thorough experiment section with discussions and insights around the results would also enhance the potential impact.
> >
> > After reading the rebuttal and other reviews, as well as re-reading the paper, I have additional comments on my concerns below. I will keep my score as the writing and motivation need substantial updates.
> >
> > **W3 [Experiments (Section 5)]:** Have the generated synthetic dataset and the MHC dataset [1] been used for online and active learning settings in the multitask regression literature before? I think that standard benchmark datasets for the specific settings should be used for empirically validating the proposed methods. Otherwise, one must clearly state in the experimental settings why and how the selected datasets apply to the online and active learning settings.
> >
> > Furthermore, details on experimental settings and datasets that could have been placed in the Appendix are missing.
> >
> > References: [1] Widmer, Christian, Nora C. Toussaint, Yasemin Altun, and Gunnar Rätsch. "Inferring latent task structure for multitask learning by multiple kernel learning." BMC bioinformatics 11, no. 8 (2010): 1-8.
> >
> > **W4 [Scope of paper]:** I believe that applying the confidence bounds to some toy dataset with deep neural networks would enhance its potential impact, and consequently its relevance, in the multitask learning community and Neurips.
> >
> > Furthermore, the abstract should be more clear on what the paper’s scope actually is to avoid misleading potential readers. The title and the abstract suggests that this paper is on general multitask learning, while it is focused on multitask kernel regression. The word ‘regression’ is not mentioned in the Abstract, which can be suggestive to the reader to believe that the paper is on general multitask learning.

---

> > > ### Author Response · Authors · 2023-08-12
> > >
> > > We thank the reviewer for going through our rebuttal.
> > > We would like to respond to the raised questions.
> > >
> > > About the considered datasets, the synthetic one was generated by us in order to evaluate the considered baselines as a function of the number of tasks $N$ and their deviation $\epsilon$ (see Figure 4 in the appendix). The MHC dataset has been typically used for multitask regression, e.g. by [1] and [2]. In both cases, offline data were used to learn a clustering structure among tasks (in [1]), or to learn a multitask kernel (in [2]). Also, it was recently utilized by [3] in the context of meta-learning. At Lines 321-324, we specify why we think this dataset is perfectly suited for *active* learning, and how we also use it for *online* learning to test our algorithms. Note that our active learning setup is novel and thus cannot be found in the existing literature.
> > >
> > > About the experimental details, we believe the definition of the synthetic dataset (Lines 315-315) and the description of the online and active learning loops (Lines 325-340) fully describe our setup. Moreover, parameters for regression and associated confidence intervals are set to their theoretical values of Theorem 1. We are also ready to release the full codebase to make our results more transparent and fully reproducible.
> > >
> > > Regarding the abstract, we precisely mention (in Line 4) that in this work we provide novel multitask confidence intervals, and that we specifically focus on online and active multitask learning. We can make more explicit that such intervals come from multitask regression; we thought that this would go without saying.
> > >
> > > About the writing style, we would like to point out that all of our sections are already introduced with “topic sentences”, see Lines 86–88 (Sec. 2), 121–122 (Sec. 2.2), 167–169 (Sec. 3), 233–238 (Sec. 3.2), 268–272 (Sec. 4), 310–312 (Sec. 5).
> > >
> > > \
> > > [1] Jacob, Laurent, et al. "Clustered multi-task learning: A convex formulation." Advances in neural information processing systems 21 (2008).
> > >
> > > [2] Widmer, Christian, et al. "Inferring latent task structure for multitask learning by multiple kernel learning." BMC bioinformatics 11.8 (2010): 1-8.
> > >
> > > [3] Rothfuss, Jonas, et al. "Pac-bayesian meta-learning: From theory to practice." arXiv preprint arXiv:2211.07206 (2022).

---

> > > > ### Comment · Reviewer_ckqH · 2023-08-14
> > > > **Reviewer Reply to Authors: Datasets for online/active learning and Presentation**
> > > >
> > > > Thank you for taking the time to respond to my questions.
> > > >
> > > > **Synthetic dataset:** It would have been helpful to see a visualization of the generated synthetic data and the model predictions. As another suggestion, a synthetic time-series dataset would have been appropriate to include for evaluating the confidence bounds in the online learning setting, e.g. see Section 3A in [1]. Even including a toy experiment of an offline multitask regression setting could be suitable as a quick sanity check of the generality of the confidence bounds.
> > > >
> > > > **MHC dataset:** I agree that your proposal of applying it in an active learning setup is novel and that you motivate it well on L321-324. However, to draw correct conclusions whether a proposed method works or not, it is important to evaluate the method on standard benchmark datasets that has been used specifically for the considered setting before. Therefore, it would have been appropriate to also include a standard benchmark dataset from the online and active learning literature that is regularly used for multitask regression tasks. In [2], under Datasets on page 4, they experiment on both synthetic and real-world datasets, where “Exam Score Prediction” and “Retail” for multitask regression tasks, that perhaps could be extended to online and active learning.
> > > >
> > > > **Experimental settings:** This part could be polished to help reproducibility, e.g. by creating bullet lists, or making a separate paragraph, with the descriptions of the baselines. The online and active learning setting for the specific datasets could also be described in the dataset description. If it doesn’t fit within the page requirement, then add the full details into the Appendix.
> > > >
> > > > **Abstract:** From reading the Abstract, it made me think that the confidence bounds are applicable for both regression and classification settings, so I think it would be appropriate to mention that this paper’s focus is on multitask regression tasks.
> > > >
> > > > **Writing style:** I agree that the use of topic sentences was done satisfyingly for the sections, and this could also be applied to the paragraphs to further improve the reading flow.
> > > >
> > > > References:
> > > >
> > > > [1] Dürichen, Robert, Marco AF Pimentel, Lei Clifton, Achim Schweikard, and David A. Clifton. "Multitask Gaussian processes for multivariate physiological time-series analysis." IEEE Transactions on Biomedical Engineering 62, no. 1 (2014): 314-322.
> > > >
> > > > [2] He, Xiao, Francesco Alesiani, and Ammar Shaker. "Efficient and scalable multi-task regression on massive number of tasks." In Proceedings of the AAAI Conference on Artificial Intelligence, vol. 33, no. 01, pp. 3763-3770. 2019.

---

> > > > > ### Author Response · Authors · 2023-08-14
> > > > >
> > > > > **Synthetic dataset**. Throughout our experiments, it was our first priority to inspect the validity of our confidence bounds. We opted for only reporting the regret achieved by the baselines because this was our ultimate quantity of interest (having valid confidence intervals is in principle only a sufficient condition to ensure small regret) and because it allowed us to average results over multiple runs. Nevertheless, we acknowledge that providing visualization of the generated tasks and associated confidence intervals can be beneficial. We have therefore focused on a single tasks’ realization, let MT-AL query tasks and pick decision points up to a given round $t$ (e.g., $t=10$ or $t=90$), and plotted the posterior mean and confidence estimates for each of the tasks given past data. We observe our improved intervals contain the unknown tasks’ functions (as predicted by theory) and are significantly less conservative than the naive ones. We are happy to add such plots to the paper.
> > > > >
> > > > > **MHC dataset**. We agree with the reviewer that it is appropriate to experiment with standard benchmarks, and we are aware of the existence of  “School Exams Data” and “Retail”, among many others. However, we explicitly chose to use the MHC-I dataset (which has also become standard in multitask learning, see e.g. [1, Section 2.7] ) because of the following reasons:
> > > > >
> > > > > - In both online and active multitask learning, the goal is to discover strategies that *maximize the performance* of each of the tasks (see definitions of regret). In MHC-I, this consists of finding the peptides with maximal binding affinity to each allele. Instead, “School Exams Data” and “Retail” data have not been collected within a performance maximization goal, but with the goal of model prediction. In “School Exams Data”, e.g., it does not make sense to iteratively “select a student” to maximize the resulting grade.
> > > > >
> > > > > - The MHC-I dataset perfectly illustrates the need for active learning, where one would like to discover the best peptides after a minimal number of trials.
> > > > >
> > > > > - The MHC-I dataset falls into our “agnostic” setting where we do not have access to any task-specific feature to enforce tasks’ similarity. Instead, in “Schools Exams Data”, e.g., school-specific features are available and could be used to learn a multitask kernel.
> > > > >
> > > > > We can make sure the above points come across clearly.
> > > > >
> > > > > We accept the rest of the suggestions to improve the paper’s flow.
> > > > >
> > > > > Finally, we would like to reiterate that our main contributions are theoretical and the main purpose of our experiments is to illustrate and validate our theory.
> > > > >
> > > > >
> > > > > \
> > > > > [1] Zhang, Yu, and Qiang Yang. "A survey on multi-task learning." IEEE Transactions on Knowledge and Data Engineering 34.12 (2021): 5586-5609.

---

### Official Review · Reviewer_kioq · 2023-07-07

**Soundness:** 3 good
**Presentation:** 3 good
**Contribution:** 3 good
**Rating:** 6
**Confidence:** 3

**Summary:**

This work studies uncertainty quantification of estimated tasks in agnostic multitask regression. Through exploiting the structure of the task Gram matrix, This work proposes an improved bound on the multitask confidence interval that is tighter than a previous confidence interval [1] from by at most $\sqrt{N}$.  With the derived interval, this work designed an online learning algorithm based on the upper confidence bound which selects the input with the largest upper confidence. This work also derives a regret bound for the online learning algorithm, showing that multitask learning is always beneficial. Moreover, this work designs a no-regret algorithm in an active learning setting, which uses confidence intervals to decide which task should be queried. This work proves that under the algorithm the active learning regret is always bounded by its online learning counterpart. To evaluate the proposed approach, this work conducts experiments on a synthetic dataset and a drug discovery dataset. In online learning, the proposed algorithm (MT-UCB) with the improved confidence interval outperforms independent learning, single-task learning, and MT-UCB with a previous bound [1]. In the setting of active learning, the proposed algorithm (MT-AL) improves over previous active learning algorithms, including MTS [2], AE-LSVI [3], and MT-AL with a previous bound [1].

 [1] Cesa-Bianchi, N., Gentile, C., and Zappella, G. (2013). A gang of bandits. *Advances in Neural*

*Information Processing Systems (NeurIPS)*.

[2] Char, I., Chung, Y., Neiswanger, W., Kandasamy, K., Nelson, A. O., Boyer, M., Kolemen, E., and Schneider, J. (2019). Offline contextual bayesian optimization. *Advances in Neural Information Processing Systems (NeurIPS)*.

[3] Li, X., Mehta, V., Kirschner, J., Char, I., Neiswanger, W., Schneider, J., Krause, A., and Bogunovic, I. (2023). Near-optimal policy identification in active reinforcement learning. *Interna- tional Conference on Learning Representations (ICRL)*.

**Strengths:**

- This paper derives an improved confidence interval that is tighter than the naive confidence interval in multitask regression.
- This paper designs two effective algorithms using the largest upper confidence based on the derived confidence interval for online multitask learning and active multitask learning. Moreover, this paper provides regret analysis for the proposed algorithm which is shown to be not worse than a previous regret bound.
- This paper empirically evaluates the proposed method in synthetic and drug discovery datasets and shows the advantages of the proposed methods in terms of regrets.

**Weaknesses:**

It would be better to provide more intuition of deriving the improved confidence interval. For example, provide a proof sketch of Theorem 1 in terms of leveraging the expression of $K_{task}(b)$. Moreover, it would be better to provide an explanation of notations from the previous work, such as $\tilde{f}$ and $\tilde{\phi}(i,x|b)$.

**Questions:**

- What would be the effect for different hyper-parameters in the algorithm, such as $b$, $N$, and $\delta$?

- What would be the interpretation for better results of AdaMT-UCB compared to MT-UCB as shown in experiments?

**Limitations:**

This work has discussed the limitations of the proposed methods.

---

> ### Author Rebuttal · Authors · 2023-08-07
>
> We thank the reviewer for the comments and suggestions. We address below the specific questions raised.
>
>
> > It would be better to provide more intuition of deriving the improved confidence interval.  For example, provide a proof sketch of Theorem 1.
>
> We agree with the reviewer’s suggestion, we expand here on the proof sketch of Theorem 1. The computation of the confidence width typically involves the inversion of some kernel-related matrix, see e.g., (Chowdhury & Gopalan 2017, Section C). In our agnostic multitask formalism, this matrix can be written as a Kronecker rank-one perturbation of an easily invertible matrix, such that one can compute the inversion in closed form, see Equation (20). We are left with two terms: one similar to independent learning, and one due to multitask learning. An appropriate bounding of the latter, depending on the value of $b$, provides the improved bound. In particular, we bound it in two different regimes: for large $b$ and for small $b$. We will add this proof sketch after Theorem 1, and recall the used notation.
>
>
> > What would be the effect for different hyper-parameters in the algorithm, such as $b$, $N$, and $\delta$?
>
> Parameter $b$ defines the agnostic multitask kernel (Eq. (2)) used for regression and intuitively controls how much information is shared across the tasks, as opposed to learning each task independently. We give further intuition on $b$ in Proposition 2 (Appendix A.1), where we show that the choice of $b$ interpolates between running $N$ independent single-task regressions ($b=0$) and treating all tasks to be identical ($b=\infty$).
>
> Parameter $N$ refers to the number of tasks, which is a problem-dependent parameter and is not tunable; it is only used by the algorithm to define the multitask kernel Eq. (2) and the associated confidence intervals (see Theorem 1). Strictly speaking, the algorithm implementation (and performance) does not require knowing $N$, but only the number of tasks observed *so far*, which is readily available information.
>
> Finally, the confidence level $\delta \in (0,1)$ is utilized to tune the width $\beta_t$ of the multitask confidence intervals, see Theorem 1. In particular, the smaller the $\delta$ the wider the confidence intervals and the higher the probability (i.e., $1-\delta$) with which they contain the true unknown task functions. In practice, $\delta$ is usually set between 0.01 and 0.05. In our experiments we conservatively chose $\delta=0.01$.
>
>
> > What would be the interpretation for better results of AdaMT-UCB compared to MT-UCB as shown in experiments?
>
> While MT-UCB requires an upper bound on the tasks’ deviation $\epsilon$ (to be used for building multitask confidence intervals), AdaMT-UCB starts with a smaller value of $\epsilon$ ($0.1$ in our experiments) and only increases it when there is evidence that the constructed intervals do not contain the true tasks. Thus, this allows AdaMT-UCB to be less conservative and potentially achieve superior performance (because the algorithm’s regret grows proportionally to the used $\epsilon$, see Theorem 2). We have remarked on this fact in our experiments section (lines 347-350). We also believe this is a very interesting finding that can be of potential independent interest, whenever one would like to combat the use of overly conservative confidence bounds.
> \
> \
> \
> We hope the above point clarified the reviewer's concerns about the paper. We are happy to expand them further if needed.

---

> > ### Comment · Reviewer_kioq · 2023-08-12
> > **Thanks for the responses!**
> >
> > I have read the authors' responses and other reviews. The responses have addressed my concerns. Therefore, I will keep my score.

---

### Official Review · Reviewer_EaHC · 2023-07-17

**Soundness:** 2 fair
**Presentation:** 2 fair
**Contribution:** 2 fair
**Rating:** 5
**Confidence:** 5

**Summary:**

This paper focuses on multitask learning in the agnostic setting. It provides multitask confidence intervals and novel online learning algorithm that achieves such improved regret. The paper also introduces a novel multitask active learning setup where several tasks must be simultaneously optimized but only one of them can be queried for feedback by the learner at each round. They design a no-regret algorithm that uses the confidence intervals to decide which task should be queried.

**Strengths:**

- Theoretical analysis in this field of MTL is valuable
- The proposed active learning problem in MTL is interesting and can be extended to other real-world applications.
- The proposed method is thoroughly evaluated on synthetic and real-world drug discovery data.

**Weaknesses:**

- Although the paper is well written, I feel that it is densely pack with multiple goals. It would be a easier to understand if the paper focus more on a single concept rather than expanding it to several applications and add empirical results for each of them.

**Questions:**

- Can you please also add other real-world applications where multi-task active learning can be extended? This can be a part of the future work section/paragraph.

**Limitations:**

- I think that the main limitation is that the proposed method isn't compared directly with existing single-task methods as baselines.
- Is it possible to have an active learning method built for single-task and apply it to all tasks in an MTL problem? How does that compare? For instance, techniques like Margin sampling or Least confidence can be easily applied to each task in an MTL problem with minimum overhead.

---

> ### Author Rebuttal · Authors · 2023-08-07
>
> We thank the reviewer for appreciating our theoretical analysis and experimental evaluations. Below we address the raised points.
>
>
> > Can you please also add other real-world applications where multi-task active learning can be extended?
>
> We agree with the reviewer’s suggestion and we will add further potential applications in the future work section. In particular, we have identified two compelling ones: *personalized health*, where one seeks to discover the best treatment for each patient, and *spam filtering*, where one seeks to obtain an accurate spam policy for each user. These are well-known applications of multitask learning but in both cases, the active learning component can play a crucial role. Indeed, our setup and approach would allow one to quickly discover the best strategy for each of the tasks (patients or email users) with minimal invasiveness.
>
> > I think that the main limitation is that the proposed method isn't compared directly with existing single-task methods as baselines.
>
> **We respectfully disagree with the reviewer.** Indeed, it is precisely the main goal of our work to show that multitask learning can be beneficial compared to single-task learning (as depicted, e.g., in Figure 1), both theoretically and experimentally as we argue below.
> In our setup, the natural single-task baseline consists of performing $N$ independent regressions, and for each task selecting actions based on the UCB rule (this is similar to Margin or Least confident sampling, but with provable no-regret guarantees). Indeed this is the main baseline we want to compare to. We denote it as ‘Independent’ in our experiments and in what follows.
> - In terms of theory, our entire analysis in Section 3 is devoted to comparing the regret of MT-UCB with the one of ‘Independent’. In particular, the first bound in Theorem 2 is the regret of ‘Independent’, while the second and third bounds refer to multitask learning. They show that multitask learning is provably beneficial whenever the number of tasks $N$ is large and the tasks’ deviation $\epsilon$ is small, as intuitively expected. To the best of our knowledge, we are the first to provably illustrate this phenomenon, and this is thanks to our refined confidence intervals derived in Theorem 1. In addition, by carefully selecting parameter $b$, our analysis also shows that MT-UCB always attains the minimum of all three bounds, hence being never worse than single-task learning.
> - In terms of experiments, we observe that ‘Independent’ is significantly and consistently outperformed by MT-UCB as expected from our theory.
>
> > Is it possible to have an active learning method built for single-task and apply it to all tasks in an MTL problem? How does that compare? For instance, techniques like Margin sampling or Least confidence can be easily applied to each task in an MTL problem with minimum overhead.
>
> The reviewer is correct that such techniques can be applied independently to each task to efficiently discover their best strategies. However, we would like to remind the reviewer that our active learning setup focuses on the challenge where *only one* of the tasks can be queried at each round. In such a case, one also needs to sequentially decide *which* task is the most informative one to query.
> \
> \
> \
> We hope the above resolve the reviewer's concerns and we are happy to provide further answers if something remains unclear.

---

### Official Review · Reviewer_J9Sp · 2023-07-27

**Soundness:** 3 good
**Presentation:** 3 good
**Contribution:** 3 good
**Rating:** 6
**Confidence:** 2

**Summary:**

The primary objective of this paper is to analyze the confidence bound of the multi-task kernel regression model and explore its application in online learning and active learning scenarios. Initially, the authors introduce a novel confidence bound, revealing distinct regimes based on task similarity. Notably, their bound proves to be tighter than a naive alternative. Leveraging this improved confidence interval, they proceed to devise online learning and active learning algorithms, providing comprehensive analyses of the confidence intervals in these contexts. In conclusion, the authors conduct empirical comparisons of their methods against existing approaches and baselines, utilizing synthetic data and drug discovery data to validate their findings.

**Strengths:**

- The analysis on the confidence interval of the multi-task kernel regression seems to be a decent addition to the related literature for adapting multi-task kernel regression to online and active learning.
- Their analysis can provide insight into how training on different tasks can be used to help out on other relevant tasks, which seems to help understanding multi-tasks learning in general.

**Weaknesses:**

I think the weakness of this work lies in the empirical results.
- The empirical results are quite limited with only one real dataset and a synthetic dataset.
- It is mentioned that for online learning, the parameter b is selected based on sweeping over all possible values and choose the best performing one. However, I am not sure if this is possible in practice. At least, a separated validation set should be used for parameter selection. (this problem seems to be also existing in the active learning setting)
- The empirical results for active learning are also a bit weak. According to Figure 3, it seems that the proposed algorithm could be slightly worse in some T and at best on par with existing algorithms. Are there any other benefits for adapting the proposed algorithm instead of other competitors in practice?
- For active leaning, I believe this work could also benefit from a more detailed analysis on the how are the examples being selected with different algorithms differs from each other.

minor:
- The y-axis in Figure 3 should have labels.

**Questions:**

- Line 217 and 218: "This suggests that the benefit of multitask may vanish with the number of available points per task, an observation which is well-known by practitioners". Are there any relevant citations?

**Limitations:**

I think the empirical results are quite limited. But this limitation is not mentioned.

---

> ### Author Rebuttal · Authors · 2023-08-07
>
> We thank the reviewer for the raised points, to which we would like to individually respond below.
> \
> \
> **W1 (empirical results on 2 datasets only).**
> We would like to point out that our work is primarily of theoretical nature and that experiments are essentially designed to support our theoretical findings. In that respect, we believe our experimental setup provides empirical evidence to all of our following claims:
> - the novel confidence intervals derived in Theorem 1 outperform their naive counterparts (as depicted in Figures 1 and 2) and are key for effective multitask learning. This is confirmed both in our online and active learning experiments where we run baselines using the naive intervals and our improved ones. In addition, in our set of synthetic experiments (see Figure 4 in Appendix D) we observe that the higher the number of tasks $N$, the more our improved intervals outperform the naive ones. This perfectly conforms with –and experimentally verifies– our theory.
> - the algorithms we introduce (MT-UCB-improved confidence, AdaMT-UCB, MT-AL-improved confidence) outperform existing methods, including the independent single-task approach, see Figures 3 and 4.
> - AdaMT-UCB, which automatically adapts to the tasks relatedness, is at least on par (if not better) with the strategy that can access (a bound on) the tasks’ deviation $\epsilon$, see Figures 3 (a) and 4.
>
> **W2 (selection of $b$).**
> Recall that we are in an online learning setting, in which the learner must make predictions at each time step for each new data point arriving. Hence, it is impossible to resort to a pre-existing validation set to select parameter $b$. We view $b$ as a tunable hyperparameter, similar to a tunable confidence width in standard Bayesian optimization (BO). However –unlike when tuning the width in BO– we note that our confidence intervals hold for any choice of $b$ (see, e.g., Figure 2), thus one could in principle employ more sophisticated hierarchical approaches (e.g. running experts on a grid of possible $b$) to perform as well as with the best $b$ in hindsight. We feel this is beyond the scope of our work and would have cluttered the paper. The situation is different for $\epsilon$, which must be a valid bound on the unknown tasks’ deviations (thus potentially affecting the validity of our confidence intervals) and for which we have designed the AdaMT-UCB adaptive procedure.
>
> **W3 (weak active learning experiments).**
> **We respectfully disagree.** First, we note that the primary goal of such experiments is to show the benefits of our improved confidence intervals over the naive ones, when utilized for active learning. In that regard, we observe that “Uniform, improved confidence” and “MT-AL, improved confidence” significantly and consistently outperform “Uniform naive” and “MT-AL naive”. Moreover, when considering the other baselines MTS and AE-LSVI, we remark that they also utilize our improved confidence intervals, which are thus partly responsible for their performance. We will make sure this comes across clearly. Second, when comparing with MTS in drug discovery, we observe that MTS has quite an aggressive behavior in the first steps, securing high rewards, but then suffers *linear* regret (i.e., it fails to converge to the optimal strategy for each task). Instead, the goal of our active learning setup is to achieve *sublinear* regret (as proved in Theorem 3), which is empirically satisfied by MT-AL in both experiments. Finally, the fact that MT-AL and AE-LSVI enjoy similar performance was pretty expected (we also mention this in Line 352). Indeed, as discussed in Appendix C3, both approaches use very similar decision rules. However, unlike for MT-AL, it is an open problem to provably bound the active learning regret of AE-LSVI.
>
> **W4 (query frequencies).**
> We have followed the reviewer’s suggestion and analyzed the frequency of each task being queried by each of the baselines. These are visible in the figures included in the attached pdf (in the general rebuttal) which we will be happy to add to the paper. In the synthetic experiments (top plot), we observe that all the active learning baselines (i.e., MT-AL, MTS, and AE-LSVI) query more often tasks 1 and 5. These are found to be the tasks with the smallest norm and consequently the smallest signal-to-noise ratio, thus requiring more learning data. In our drug-discovery experiments, unfortunately, we cannot compute such norms since we do not have access to the underlying task vectors. However, we notice that MT-AL and AE-LSVI query with slightly more frequency task-3 (corresponding to allele A-0203). Instead, MTS focuses heavily on task-1 due to its different query strategy. Indeed, unlike MT-AL and AE-LSVI which query the task with maximal uncertainty, MTS queries the task with maximal potential performance increase. This aggressive strategy may explain the observed imbalance and the regret of MTS which grows linearly (Figure 3, rightmost plot).
>
> **Minor weakness (Figure 3).** As specified in the caption of the figure, the y-axes in Figure 3 represent the regrets (online and active). We will add a label in the revision of the paper.
>
> **Question (missing reference).** We will add relevant citations. See for instance Section 5.4 in “When is Multi-task Learning Beneficial for Low-Resource Noisy Code-switched User-generated Algerian Texts?” by Adouane and Bernardy (2020).
> \
> \
> \
> We hope the above points clarified the reviewer’s doubts about our work. We are happy to provide more detailed answers based on the reviewer's feedback.

---

### Author Rebuttal · Authors · 2023-08-07

We thank all the reviewers for going through our work and providing insightful suggestions and comments.
We have addressed each of the reviews individually, hoping that most of the reviewer’s concerns are resolved.

We are happy to provide additional insights and details based on the reviewers’ feedback.

The attached *.pdf refers to the additional details inquired by Reviewer J9Sp.

---

### Decision · Program_Chairs · 2023-09-21

**Decision:**

Accept (poster)

**Comment:**

Based on the reviews and the responses, the reviewers are in broad agreement that:

- The paper makes a solid theoretical contribution on constructing confidence intervals for multi-task kernel regression
- This paper empirically evaluates the proposed method in synthetic and drug discovery datasets

In the discussion phase, Reviewer w9gE and Reviewer J9Sp championed this paper due to its solid theoretical contributions. However they also acknowledged Reviewer ckqH's presentation suggestion that the paper should make it clear from the title and the abstract that the paper focuses on an online multitask kernel regression setting, not a general multitask learning setting. Please take this into account in the camera-ready revision.